# On the presence of high nitrite ($NO_2^-$) in coarse particles at Mt. Qomolangma

Zhongyi Zhang[1, 2], Chunxiang Ye[3], Yichao Wu[1], Tao Zhou[4], Pengfei Chen[5,6], Shichang Kang[5,6], Chong Zhang[3], Zhuang Jiang[1], Lei Geng[1, 2, 4*]

[1]Deep Space Exploration Laboratory/School of Earth and Space Sciences, University of Science and Technology of China, Hefei 230026, Anhui, China

[2]CAS Center for Excellence in Comparative Planetology, University of Science and Technology of China, Hefei 230026, Anhui, China

[3]SKL-ESPC & SEPKL-AERM, College of Environmental Sciences and Engineering, and Center for Environment and Science, Peking University, Beijing 100871, China

[4]National Key Laboratory of Deep Space Exploration, Hefei, 230088, Anhui, China.

[5]Key Laboratory of Cryospheric Science and Frozen Soil Engineering, Northwest Institute of Eco-Environment and Resources, Chinese Academy of Sciences; Lanzhou, 730000, Gansu, China

[6]University of Chinese Academy of Sciences; Beijing, 100049, China

* *Correspondence to*: Lei Geng (Email: genglei@ustc.edu.cn; Tel: +86-0551-63600015)

## Abstract

Atmospheric reactive nitrogen cycling, with nitrous acid (HONO) and particulate nitrite ($NO_2^-$) as important intermediates, is crucial for maintaining the atmospheric oxidation capacity of background atmosphere on the Tibetan Plateau. During an eleven-day field campaign at the Base Camp of Mt. Qomolangma in spring of 2022, we observed significant enrichments of $NO_2^-$ in total suspended particulate (TSP) with a mean concentration of $375 \pm 386$ ng m$^{-3}$, while $NO_2^-$ was absent in fine particles (PM$_{2.5}$). The comparison revealed that $NO_2^-$ predominately exists in coarse particles. Local surface soil at the sampling site also exhibited high levels of $NO_2^-$, with $\delta^{15}N$ value similar to $NO_2^-$ in TSP. This isotopic similarity suggests that wind-blown soil is probably the primary source of $NO_2^-$ in TSP, accounting for the background levels. While concentration changes of water-soluble inorganic ions in TSP and PM$_{2.5}$ in response to shifts in air mass back-trajectories imply that atmospheric pollutants transported from South Asia may further elevate the $NO_2^-$, the specific mechanisms of long-range transport resulting in $NO_2^-$ accumulation in TSP rather than PM$_{2.5}$ remain unknown and need to be investigated. The elevated levels of TSP $NO_2^-$ may readily participate in atmospheric reactive nitrogen cycling through gas-particle partitioning or photolysis, leading to the production of HONO, OH and NO and thereby influencing oxidation chemistry. Further efforts on the sources and atmospheric chemistry of particulate nitrite are warranted, particularly in the pristine Tibetan Plateau, where even small inputs of $NO_x$ or HONO can disproportionately affect oxidant budgets and reactive nitrogen cycling.

## 1 Introduction

The Himalayas-Tibetan Plateau (HTP) represents one of the most important geomorphologically cryospheric regions, boasting high abundance of alpine glaciers and extensive areas of high-altitude snow cover (Kehrwald et al., 2008; Yao et al., 2012; Kang et al., 2019). The melting of snow cover and glacier ice holds immense importance as crucial freshwater resources for over 1.4 billion people in Asia, earning HTP the title as the "Water Tower of Asia" (Immerzeel et al., 2010). However, the enhanced warming rate in the Tibetan Plateau (TP, ~ 0.30°C per decade) has resulted in a rapid and alarming increase in glacier melting over the past decade, significantly affecting the climate, hydrological cycles, and ecosystems at local and global scales (Immerzeel et

al., 2010; Xu et al., 2009; Lau et al., 2010). Persistently increased aerosol loadings and greenhouse gas in TP region account for the increased warming rate (Kang et al., 2019; Lau et al., 2010; Lüthi et al., 2015). More importantly, once deposited on the surface of snow cover and glacier ice, the aerosol, especially these light-absorbing components (i.e., black carbon, dust) contribute significantly to the rapid glacial retreat (Xu et al., 2009; Zhao et al., 2020).

Atmospheric oxidation capacity (AOC) regulates secondary aerosol formation and trace gases removal, including $CH_4$ (Wang et al., 2023; Ye et al., 2023; Ye et al., 2016; Andersen et al., 2023), therefore acting as a critical link between atmospheric pollution and cryospheric changes. Previous study have suggested strong solar radiation, high $O_3$ and relatively high water vapor dominate the relatively strong AOC over the TP (Lin et al., 2008). Recent field campaign further highlighted the rapid reactive nitrogen cycling, with N(III) species (i.e., HONO) as the intermediate, also plays an important role in maintaining the strong AOC in TP (Wang et al., 2023). For example, *Wang et al.* reported high-than-expected HONO (~30 ± 13 pptv) in the Namco station, a typical background site in the middle of TP, with HONO sources including $NO_2$ heterogenous conversion, soil emission and particulate nitrate photolysis (Wang et al., 2023). However, a detailed HONO budget analysis indicated these three dominant sources could not account for the observed daytime HONO levels at the background site, implying the existence of additional, yet unidentified, HONO sources. Particulate nitrite ($NO_2^-$) likely represents a potential source of HONO through thermodynamic partitioning processes under favorable atmospheric conditions, provided particulate nitrite is present in significant amounts (Vandenboer et al., 2014a; Chen et al., 2019; Li, 1994). Interestingly, relatively high levels of nitrite in total suspended particulate (TSP) have also been reported from remote sites of TP, i.e., in a forest site in the Southeast Tibet (~ 140 ng m$^{-3}$) and at the Qomolangma monitoring station (QOMS, ~ 60 ng m$^{-3}$) (Bhattarai et al., 2019; Bhattarai et al., 2023). Such high levels of particulate $NO_2^-$ may also contribute to the strong AOC in TP, either via directly photolysis to produce $NO_x$ (Jacobi et al., 2014) or indirectly serve as an important source of HONO through gas-particle partitioning (R1). However, the sources and formation mechanisms for the relatively high level of atmospheric $NO_2^-$ observed in the TP remain unclear.

$$HONO_{(g)} \leftrightarrow HONO_{(aq)} \leftrightarrow NO_2^-{}_{(aq)} + H^+{}_{(aq)} \text{ (R1)}$$

$$HONO_{(aq)} + H^+{}_{(aq)} \leftrightarrow H_2ONO^+{}_{(aq)} \text{ (R2)}$$

$$HONO + h\upsilon \ (300 \ nm < \lambda < 405 \ nm) \rightarrow NO + OH \quad (R3)$$
The stable nitrogen and oxygen isotopic compositions ($\delta^{15}N$, $\delta^{18}O$, and $\Delta^{17}O$; where $\delta =$
$(R_{sample}/R_{reference} - 1) \times 1000‰$ and with $R$ denoting the $^{15}N/^{14}N$, $^{18}O/^{16}O$, and $^{17}O/^{16}O$ ratios; $\Delta^{17}O =$
$\delta^{17}O - 0.52 \times \delta^{18}O$) may provide diagnostic information regarding the sources and formation
pathways of atmospheric nitrite. Similar isotopic approaches have been widely used to explore the
nitrate ($NO_3^-$) sources (Morin et al., 2008; Zong et al., 2020; Geng et al., 2014; Fang et al., 2011;
Hastings et al., 2003; Zhang et al., 2022; Zhang et al., 2021b; Liu et al., 2018; Felix and Elliott,
2014; Miller et al., 2018). Considering that atmospheric $NO_2^-$ may share similar sources and
formation pathways with $NO_3^-$, the specific $NO_2^-$ formation pathways are expected to be
characterized by distinct oxygen or nitrogen isotopic endmembers, despite reports on the
atmospheric $NO_2^-$ isotopic compositions are rare. For instance, $NO_2^-$ produced from the photolysis
of particulate $NO_3^-$ may possess very negative $\delta^{15}N$ values compared to $NO_3^-$, analogous to the
pronounced nitrogen isotope fractionation effects associated with snow nitrate photolysis (Erbland
et al., 2013; Frey et al., 2009), while the $\Delta^{17}O$ of $NO_2^-$ is expected to closely resemble that of $NO_3^-$
as the oxygen atom in $NO_2^-$ is imparted from $NO_3^-$, unless significant oxygen atom exchange
between $NO_2^-$ and aerosol water occurs. The $\Delta^{17}O$ of $NO_2^-$ (and HONO) from primary emission
sources is expected to be negligible, while that generated from heterogeneous reactions of $NO_2$ on
the aerosol surface would be characterized by positive $\Delta^{17}O$ values depending on the degree of $NO_2^-$
and aerosol water oxygen isotope exchange. These unique isotopic fingerprints may be utilized in
distinguishing the sources and formation pathways of atmospheric $NO_2^-$.
To gain insight into the sources and/or formation mechanisms of atmospheric $NO_2^-$ in the pristine
environment of TP, we collected the TSP and fine particulate matter ($PM_{2.5}$) synchronously at the
Base Camp, the north slope of the Mt. Qomolangma during the campaign of "Earth Summit
Mission-2022" scientific expedition from April 24[th] to May 6[th], 2022, with additional surface soil
samples collected in May, 2023. The $NO_2^-$ concentration and multi-isotopic signatures ($\delta^{15}N$, $\delta^{18}O$,
and $\Delta^{17}O$) in aerosol and surface soil were then determined in order to evaluate the potential sources
of atmospheric $NO_2^-$. Additionally, the potential environmental implication of atmospheric nitrite
was explored in the term of atmospheric oxidation capacity at this pristine environment.

## 2 Material and Method

### 2.1 Site description

The Base Camp is located in the middle of the Rongbuk valley (86.85 °E, 28.14 °N), situated ~5200 m above sea level (m a.s.l) on the north slope of the Mt. Qomolangma (Zou et al., 2008; Zhu et al., 2006). The surrounding surface consists of loosed soil, gravel, broken rocks of various sizes, with sparse vegetation due to the semi-arid status (Ming et al., 2007; Zou et al., 2008). Rongbuk valley is characterized by a depth of ~1000 m and a floor width of ~1000 m, with elevations of the surrounding mountains exceeding 6000 m on both sides (Zou et al., 2008). Attributed to the unique topography, the local air circulation is dominated by mountain and valley breezes. The predominant wind regime is the katabatic flow of southerly and southeasterly, which is typically persists from noon to midnight (Zhu et al., 2006; Zou et al., 2008; Zhou et al., 2011). The nearest accessible area for residents and visitors is at least 2 km north of the Base Camp. During the campaign, electricity and natural gases were routinely used for cooking and hot water production. There were intermittent vehicle exhaust emissions around the station during daytime for the daily necessaries supporting, i.e., water and food. To minimize the influence of local anthropogenic activities on sampling, the instruments were set in the southeast (upwind direction, Figure S1) and approximately 100 m away from the living space of the Base Camp. The anthropogenic influence on the sampling is expected to be minimal.

### 2.2 Field campaign and sample collection

From April 26[th] to May 6[th], 2022, TSP samples were collected simultaneously with $NO_2$ using a homemade denuder-filter system (Zhou et al., 2022). A similar system has been widely used to separately collect atmospheric particulate matter and N-containing gases for isotopic analysis (Chai et al., 2019; Chai et al., 2021), suggesting the mutual interference between the particulate and gaseous phases to be minimal. A polytetrafluoroethylene (PTFE) sleeve is used to assemble the homemade denuder with the filter pack, flowmeter, and pump. The filter pack was placed in the front of the denuder. All connections between the various parts of the sampling apparatus are made using 3/8″ Teflon tubing. A detailed description of the sampling apparatus can be found in our previous report (Zhou et al., 2022). Whatman quartz filter (circles, diam. 47mm, pre-heated at 400 ℃

for 3 h before use) was placed into the filter pack to collect TSP sample. In the present study, the
collected bulk aerosol can be regarded as TSP since no size-selective inlet was employed. The flow
rate was controlled at 30 Lmin$^{-1}$ using a flowmeter. Previous reports have indicated that the flow
rate has negligible effect on the concentrations of sulphate, nitrate, and ammonium when using
quartz filter for sampling (Keck and Wittmaack, 2005). To minimize the potential influence of the
loose ground surface on the TSP collection, a mountain tent was used to separate the pump (out the
tent) with the denuder-filter system (in the other side of tent), and the inlet Teflon tube was stretched
out of the tent for ~1.5 m height straight.
From April 24$^{th}$ to May 6$^{th}$, 2022, PM$_{2.5}$ were sampled using a high-volume aerosol sampler
(TH-1000F; Wuhan Tianhong Instruments Co. Ltd., China) equipped with PM$_{2.5}$ inlet and Whatman
quartz-fiber filters (sheets, 203 mm × 254 mm) at a flow rate of 1.5 m$^3$/min. All the quartz filters
were pre-heated at 400 ℃ for 3 h before use. In general, TSP and PM$_{2.5}$ samples were collected with
diurnal resolution during this campaign, with daytime samples from approximately 09:00–20:00
and nighttime samples from 21:00–08:00 (local time), respectively. From May 2$^{nd}$ to May 4$^{th}$, we
collected the daytime TSP and PM$_{2.5}$ samples in the morning (09:00-14:00) and afternoon (14:00-
20:00), respectively. A total of 24 TSP samples (including 2 blanks) and 29 PM$_{2.5}$ (including 3 blanks)
were collected during this campaign. After each sampling period, quartz filters were removed from
the filter pack/PM$_{2.5}$ inlet and immediately wrapped in pre-baked aluminum foil and then stored in
frozen until analysis to minimize potential loss of nitrite. During the campaign, snowfall events
occurred on the night of April 29$^{th}$ and during the daytime of April 30$^{th}$.
Surface soil samples (0-5 cm depth, n = 9) were collected in May, 2023 from the east slope,
west slope and south sides of the Rongbuk valley. A polytetrafluoroethylene (PTFE) shovel was
used to collect soil. The collected soil was immediately transferred to clean plastic bags, sealed and
kept frozen. Soil samples were transported into laboratory using a cold chain. Upon arrival at our
laboratory, the soil samples were passed through a 60-mesh screen (~0.25 mm) to remove larger
particles and thoroughly homogenized prior to chemical and isotopic analysis.
**2.3 Ionic concentration analysis and uncertainty estimation**
Water-soluble inorganic ions (WSIs, including Na$^+$, NH$_4^+$, K$^+$, Mg$^{2+}$, Ca$^{2+}$, Cl$^-$, NO$_2^-$, NO$_3^-$, and
$SO_4^{2-}$) in TSP (entire filter, ~16.6 cm$^2$) and PM$_{2.5}$ (1/32 section, ~13.0 cm$^2$) were extracted using 20
mL Milli-Q ultrapure water (18.2Ω cm) in an ultrasonic bath at room temperature for 30 min. Note
the TSP filters were cut to fit the filter holder from the standard Whatman quartz-fiber filters which
were also used as the PM$_{2.5}$ filters. After filtration through a 0.22 μm pore size syringe filter which
was pre-cleaned with ultrapure water, the filtrate was subjected to inorganic species analysis using
ion chromatography (Dionex Aquion) (Zhang et al., 2020). Blank filters were pretreated and
measured the same as real samples, and the limits of detection (LOD) was calculated as 3 times of
standard deviations of blanks (Fang et al., 2015).  In general, Na$^+$ in the blank filters is comparable
to samples, a well-known issue for the Whatman quartz filter which is high in Na$^+$ blank. Therefore,
in this study, we discarded the Na$^+$ data. The volatile components (i.e., NH$_4^+$, NO$_2^-$, NO$_3^-$) and K$^+$
in blank are low but several times higher than the detection limits; SO$_4^{2-}$, Mg$^{2+}$ and Ca$^{2+}$ in blank
are significantly higher than the respective LOD but lower than samples by at least five times (the
lower end). All reported concentrations of each ion were blank corrected as follows:
$$C_i = \frac{(\rho_{sample} - \rho_{blank}) \times V_{water} \times F}{V_{air}}$$

with $C_i$ representing the ambient concentrations of specie $i$ in air (ng m$^{-3}$ or μg m$^{-3}$), $\rho_{sample}$ and
$\rho_{blank}$ are the concentrations determined by the ion chromatography (ng mL$^{-1}$), $V_{water}$ is the volume
of ultrapure water used for extraction (20 mL), $V_{air}$ is the volume of air sampled for each PM$_{2.5}$ or
TSP filter, F is the ratio of particulate matter collection area for PM$_{2.5}$ or TSP filters to the filter area
used for extraction.
The overall uncertainty in ion concentration was estimated according to the law of error
propagation, accounting for the sampling air volume (3% for PM$_{2.5}$ samples and 1% for TSP samples
as provided by the manufactures), the extraction of water volume (~0.3% for pipetting from the
manufacture Eppendorf), the blanks, and the analytical uncertainty from ion chromatography and
calibration, assuming that these factors are independent. The analytical uncertainty for water-soluble
ions concentration determination using ionic chromatography has been extensively assessed in our
laboratory, with values typically <5% for all inorganic species at concentration of 500 ng mL$^{-1}$. The
combined uncertainty about the ionic concentrations in PM$_{2.5}$ and TSP are shown in Table S1. In
general, TSP samples are associated with relatively high overall uncertainty compared to PM$_{2.5}$
samples, perhaps due to the relatively high blank variability due to the low mass loading in TSP.
For soil ionic concentration analysis, 4.0 g sieved soil was extracted using 20 mL Milli-Q
ultrapure water, shaken for 30min at room temperature. After centrifugation, the extract was passed
through 0.45 mm filters before ions analysis. The concentration determination of water-soluble
inorganic ions in soil extract was similar to that of aerosol samples. Detailed procedure for the
extraction of soil inorganic ions, especially the $NO_2^-$ has been descried in previous report (Homyak
et al., 2015).
**2.4 Isotopic analysis**
After concentration measurements, isotopic analyses ($\delta^{15}N$, $\delta^{18}O$, and $\Delta^{17}O$) of $NO_2^-$ in TSP
were conducted using the azide method (Casciotti et al., 2007). The azide method can reduce nitrite
ion in solution into $N_2O$ in a single step, while nitrate ion remains unchanged, ensuring no
interference on the nitrite isotopic analysis. The azide reagent is prepared by mixing 2 M sodium
azide with 40% acetic acid at a 1:1 ratio by volume in our laboratory. $NO_2^-$ standards and samples
were pretreated under identical conditions concerning the total volume, nitrite amount, water isotope,
and matrix. The $\delta^{15}N$, $\delta^{18}O$, and $\delta^{17}O$ of $N_2O$ reduced from $NO_2^-$ in the TSP samples and standards
were determined using a Finnigan® MAT253 plus isotope ratio mass spectrometer (IRMS) equipped
with a GasBench II and preconcentration system. The data calibration followed the procedures
described in *Albertin et al*., 2021, using three international $KNO_2$ salt standards (RSIL-N10219,
RSIL-N7373, and RSIL-N23 with respective $\delta^{15}N$ and $\delta^{18}O$ values of 2.8/88.5 ‰, -79.6/4.2 ‰, and
3.7/11.4‰). The $\Delta^{17}O$ values of the three international references have not been certified. To address
this, a series laboratory experiments was conducted to determine the true values of three
international references in our laboratory (Zhang et al., 2025). In brief, each nitrite international
reference was oxidized into $NO_3^-$ by $O_3$ produced from commercial ozone generator. A parallel flow
of $O_3$ was also used to convert a normal $KNO_2$ salt ($\Delta^{17}O = 0$) into $NO_3^-$ to quantify the $\Delta^{17}O$ transfer
during $O_3$ oxidation, following the approach of (Vicars and Savarino, 2014). Based on these
experiments, the $\Delta^{17}O$ of RSIL-N7373 and RSIL-N23 are determined to be negligible, consistent
with previous findings (Albertin et al., 2021), while the $\Delta^{17}O$ of RSIL-N10219 is determined to be
(-9.3 ± 0.2) ‰ in our laboratory. The $^{17}O$-excess in samples is then calculated as $\Delta^{17}O = \delta^{18}O - 0.52$
$\times \delta^{17}O$. The standard deviations for $\delta^{15}N$, $\delta^{18}O$, $\Delta^{17}O$ of reference materials (n = 10) were determined
to be less than 0.1‰, 0.6‰, and 0.4‰, respectively.
For soil $NO_2^-$ isotopic analysis, ~30.0g sieved soil was extracted using 150 mL Milli-Q
purewater. The soil extract was then preconcentrated into 10 mL using ion-exchange resin before
isotopic analysis. The preconcentration approach was widely used for nitrate isotopic analysis in
snow and ice samples, and the detailed procedures and the performance have been provided in
*Erbland et al.,* 2013 and evaluated in our laboratory (Text S1). The concentrated soil extracts (50
nmol $NO_2^-$) was then subjected to soil $NO_2^-$ isotopic analysis by converting into $N_2O$ via the azide
method. The remaining procedures of soil $NO_3^-$ and $NO_2^-$ analysis were same as those for TSP
samples, as aforementioned.
**2.5 Complementary analyses of air mass backward trajectory**
To evaluate the possible impact of biomass burning emissions or other pollution sources from
South Asia, the Hybrid Single-Particle Lagrangian Integrated Trajectory (HYSPLIT) model
(performed using TrajStat plugin of the MeteoInfo software) and archived Global Data Assimilation
System (GDAS) of meteorological data were used to model the air mass back trajectories (Wang,
2014). In this study, 3-day air mass backward trajectories with arriving height of 1000 m above
ground level were simulated to identify the most likely pathway and potential source regions of the
air masses at the Base Camp (Bhattarai et al., 2023; Lin et al., 2021). Moreover, the Fire Information
and Resource Management System (FIRMS) developed by Moderate Resolution Imaging
Spectrometer (MODIS) (https://worldview.earthdata.nasa.gov) was employed to identify the
distribution of active fire spots during the sampling period.
**3   Results**
**3.1 Mass concentrations of water-soluble inorganic ions in TSP and PM$_{2.5}$**
Figure 1 displays the chemical compositions of water-soluble inorganic ions, their
corresponding time series and fractional contributions in TSP and PM$_{2.5}$. Throughout the campaign,
substantial variations of total WSIs in PM$_{2.5}$ and TSP were observed. For PM$_{2.5}$ samples, the mass
concentrations of total WSIs before April 30[th] were higher than that from May 1[st] to May 6[th] (4.1 $\pm$
1.7 versus 1.7 $\pm$ 0.6 $\mu g\ m^{-3}$; $p < 0.05$). The cut-off date of April 30[th] was selected based on observed
shifts in concentrations of water-soluble ions and air mass origins (described in section 4.2). This
decline of total WSIs after April 30th was predominately driven by significant reductions in
secondary inorganic species, i.e., $SO_4^{2-}$, $NO_3^-$ and $NH_4^+$, with the magnitude by more than 60%. In
particular, $NH_4^+$ in $PM_{2.5}$ was on average $(322 \pm 243)$ ng m$^{-3}$ before April 30th whereas $NH_4^+$ in $PM_{2.5}$
collected during daytime and nighttime of May 1st were 1 ng m$^{-3}$ and 3 ng m$^{-3}$, respectively, and
$NH_4^+$ in $PM_{2.5}$ extractions from May 2nd to May 6th was below the detection limit. Therefore, the
fractional contribution of secondary inorganic species in $PM_{2.5}$ also decreased (Figure 1d). Similarly,
$K^+$ in $PM_{2.5}$, a good tracer of biomass burning (Ma et al., 2003), also declined significantly after
April 30th $(269 \pm 432$ versus $22 \pm 12$ ng m$^{-3}$; $p < 0.05$). The elevated concentrations of WSIs before
April 30th $(4.1\pm1.7\mu g$ m$^{-3})$ are comparable to previous reports at QOMS station $(4.2 \pm 2.2$ μg m$^{-3})$
in the spring (Lin et al., 2021). In comparison, concentrations of $Ca^{2+}$ and $Mg^{2+}$, tracers of wind-
blown dust (Wang et al., 2002), decreased by less than 20% after April 30th. In general, $SO_4^{2-}$, $NO_3^-$,
and $Ca^{2+}$ are the most abundant species in $PM_{2.5}$, accounting for the majority of the mass of total
WSIs. In addition, no clear diurnal variation of water-soluble inorganic ions in $PM_{2.5}$ was observed
in this study (Figure S2).

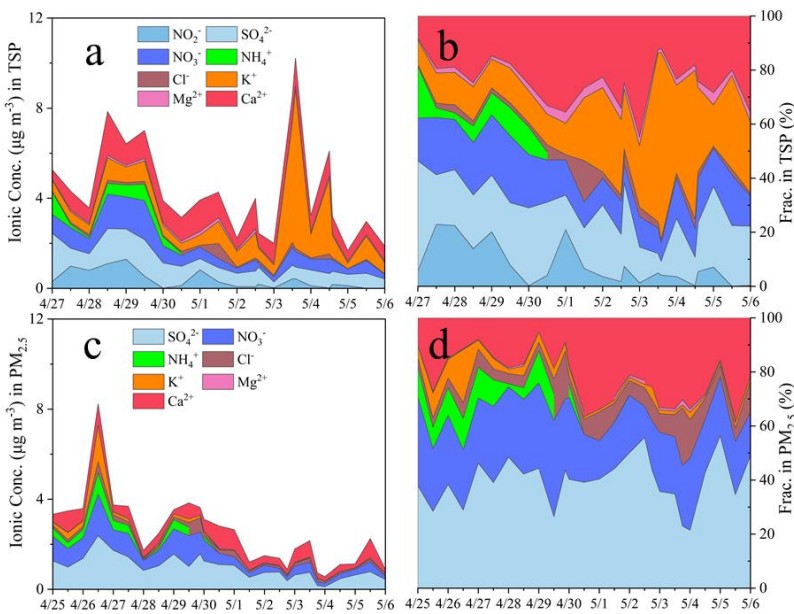


**Figure 1.** The chemical compositions and time series of mass concentrations of water-soluble
inorganic species ($NO_2^-$, $SO_4^{2-}$, $NO_3^-$, $Ca^{2+}$, etc.), as well as the corresponding mass fractions in
respective TSP (a, b) and $PM_{2.5}$ (c, d) samples collected at Base Camp of Mt. Qomolangma in spring
266    2022.

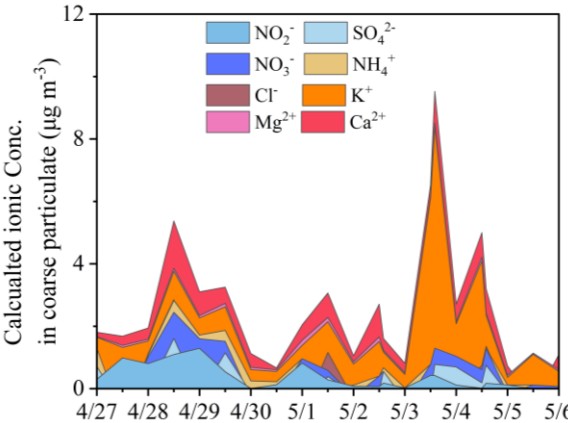


**Figure 2.** The estimated chemical compositions and time series of mass concentrations of chemical species ($NO_2^-$, $SO_4^{2-}$, $NO_3^-$, $Ca^{2+}$, etc.) in coarse-mode particulate during the springtime campaign (calculated as the difference between TSP and $PM_{2.5}$). Discontinuities in the time series were observed for certain species (e.g., $SO_4^{2-}$), likely resulting from relatively lower concentrations in the TSP samples compared to those in the corresponding $PM_{2.5}$ samples. This discrepancy is likely attributed to the propagated uncertainties involved in the concentration analysis, sampling approach and blank corrections.

Some water-soluble inorganic ions (e.g., $SO_4^{2-}$, $NO_3^-$ and $NH_4^+$) in TSP showed similar variation trends with that in $PM_{2.5}$ throughout the campaign, while others, such as $K^+$, exhibited divergent behavior (Figure 1). For example, after April 30th, the secondary inorganic species in TSP declined considerably by over 50% (i.e., $NH_4^+$ in TSP declined by more than tenfold), while $Ca^{2+}$ in TSP decreased with a smaller degree (< 15%) and $Mg^{2+}$ remained stable. In contrast, TSP $K^+$ (from both crustal and biomass burning sources)(Hsu et al., 2009) drastically surged on May 3rd and May 4th. Note that other species in TSP, i.e., $SO_4^{2-}$ and $NO_3^-$ also increased to some extent on May 3rd. The time series of water-soluble inorganic ions in the coarse-mode particulate, calculated as the differences between TSP and $PM_{2.5}$, are presented in Figure 2. Due to the overall analytical uncertainties, the air concentrations of $SO_4^{2-}$ in $PM_{2.5}$ samples occasionally exceeded that in corresponding TSP samples on some days.

Figure 3 presents the ion balance of measured water-soluble ions in $PM_{2.5}$ throughout the campaign as well as in TSP collected before and after April 30th, respectively, to highlight the significant decline in TSP $NO_2^-$. There is a strong correlation between cation and anion equivalents in $PM_{2.5}$ samples ($R^2 = 0.70$), whereas the correlations decreased in TSP samples ($R^2 = 0.46$ before April 30th and 0.49 after that, respectively). The nanogram-equivalent weight of cation were

significantly higher than that of anions for all samples, with ratio of cation to anion equivalent of $\sim$ 1.5 for $PM_{2.5}$, ~1.9 for TSP collected before April 30[th], ~4.2 for TSP collected after May 1[st]. Clearly, the declines in TSP $NO_2^-$ after April 30[th] resulted in relatively higher cation/anion equivalent ratio. The slopes of the correlation lines exceeded unity for $PM_{2.5}$ and TSP samples, indicating the alkaline nature of aerosol. The observed deficiency of anion can be attributed to the presence of carbonates (i.e., $CaCO_3$), which can dissolve in water during extraction to release $CO_3^{2-}$ and/or $HCO_3^-$ despite the relatively low solubility (Zhang et al., 2021a).

The most distinct feature of chemical compositions in TSP was the elevated level and significant variation of $NO_2^-$, ranging from 0.2 ng m$^{-3}$ to 1291 ng m$^{-3}$ in air and with an average of 375 $\pm$ 386 ng m$^{-3}$. In comparison, $NO_2^-$ consistently remained below the detection limit in $PM_{2.5}$ samples. Note during the laboratory measurements of ionic concentrations, TSP and $PM_{2.5}$ filters were extracted with ultrapure water and it was the extraction analyzed by ion chromatography. To ensure fair comparisons, similar areas of filters were extracted with same volume of ultrapure water, so that the extractions from the $PM_{2.5}$ filters should be more concentrated in atmospheric particulate species compared to that from the TSP filters, since $PM_{2.5}$ samples were collected at a much faster sampling speed (1.5 m$^3$ min$^{-1}$ vs. 30 L min$^{-1}$) over the same sampling duration. Nevertheless, $NO_2^-$ was detectable only in the extractions of TSP filter. The determined $NO_2^-$ concentrations in TSP in this study (375 $\pm$ 386 ng m$^{-3}$) were higher than previous reports conducted in various remote sites, such as at QOMS station (~60 ng m$^{-3}$ for TSP) (Bhattarai et al., 2023), at a forest site in the Southeast Tibet Plateau (~140 ng m$^{-3}$ for TSP) (Bhattarai et al., 2019), in the middle hills of the central Himalayas (~210 ng m$^{-3}$ for TSP) (Tripathee et al., 2021).

In particular, there was a dramatic decrease in TSP $NO_2^-$ after April 30[th], from a mean of (625 $\pm$ 457) ng m$^{-3}$ to (147 $\pm$ 145) ng m$^{-3}$, in line with the declines in other secondary inorganic species. Over the course of the campaign, $NO_2^-$ comprised approximately 8% of the total WSIs mass in TSP, while its contribution reached maximum of ~20% on April 27[th] and April 28[th], being one of the most abundant components on the two days. In addition, there was a strong correlation between $NO_2^-$ and $NO_3^-$ throughout the campaign ($r = 0.75$, $p < 0.05$. Figure S3). Meanwhile, the mean mass ratio of $NO_2^-$ to $NO_3^-$ was ~50% throughout the campaign, but on several days (i.e., night of April 27[th]) $NO_2^-$ concentrations significantly exceeded that of $NO_3^-$ (802 vs. 663 ng m$^{-3}$). Previous study also reported

320 comparable $NO_2^-$ and $NO_3^-$ concentrations in at a forest site in the Southeast Tibet (summer:100 ng

321 $m^{-3}$ vs. 110 ng $m^{-3}$; winter: 180 ng $m^{-3}$ vs. 270 ng $m^{-3}$) (Bhattarai et al., 2019).

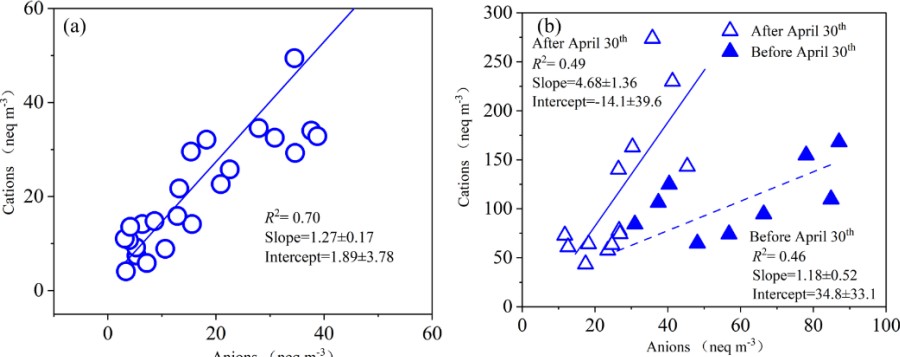

322

**Figure 3.** Ion balance for $PM_{2.5}$ (a, blue circles) and TSP samples (b, blue hollow triangles: samples collected after April 30th; blue solid triangles: samples collected before April 30th). Concentrations are expressed in nanogram-equivalent weight per cubic meter (neq $m^{-3}$).

## 3.2 Isotopic signatures of nitrite in TSP

Figure 4 presents the times series of $\delta^{15}N$, $\delta^{18}O$ and $\Delta^{17}O$ of $NO_2^-$ in TSP, along with the $NO_2^-$
concentrations. Similar to the variation trend of $NO_2^-$ concentrations, $NO_2^-$ isotopes varied in a wider
range before April 30th, but became more stable afterward. For example, $\delta^{15}N(NO_2^-)$ ranged from -
10.9 ‰ to 0.8 ‰, with a relatively large standard deviation before April 30th compared to that after
May 1st ((-6.4 ± 4.3) ‰ vs. (-8.0 ± 0.7) ‰). The large variability in $\delta^{15}N(NO_2^-)$ before May 1st is
predominately attributed to the two high values observed in daytime of April 27th and night of April
28th. Note that the two high $\delta^{15}N(NO_2^-)$ samples were also associated with relatively high $NO_2^-$ mass
concentrations. In contrast to the declining trend of $NO_2^-$ concentrations, the mean $\delta^{15}N(NO_2^-)$
values were comparable before and after April 30th. Relatively large variability was observed in TSP
$\delta^{18}O(NO_2^-)$, ranging from -9.0‰ to 3.9‰ and with an average of (-3.4 ± 3.8) ‰. TSP $\Delta^{17}O(NO_2^-)$
varied within a narrow range from -0.2‰ to +0.7‰ and with a mean of (0.2 ± 0.3) ‰. During the
campaign, no significant correlations were observed between $\delta^{15}N(NO_2^-)$ and the $NO_2^-$
concentrations; while $\delta^{18}O(NO_2^-)$ appeared to be moderately correlated with the $NO_2^-$
concentrations (Figure S5).

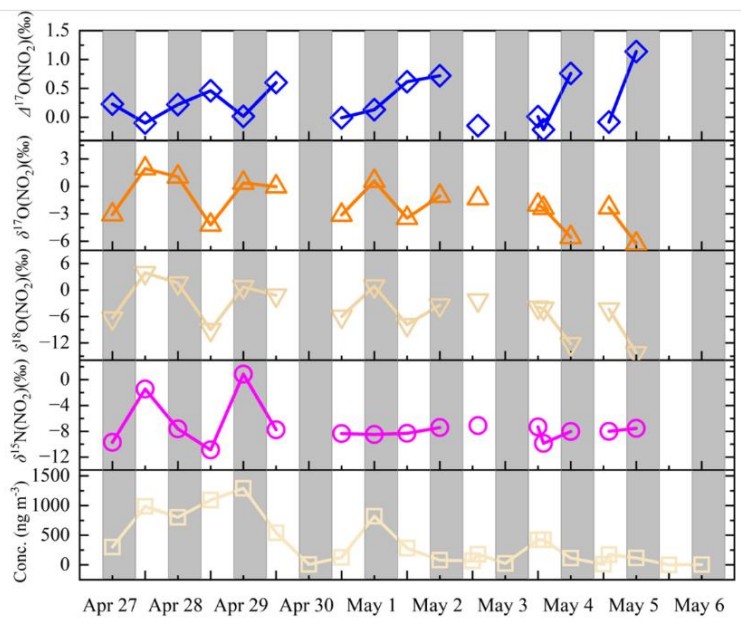

**Figure 4.** Time series of multiple isotopic signatures of $NO_2^-$ ($\delta^{15}N$, $\delta^{18}O$, $\delta^{17}O$ and $\Delta^{17}O$) as well as
corresponding concentrations in TSP samples during "Earth Summit Mission" scientific expedition
in spring 2022. The gray shaded area denotes local nighttime.
**Table 1.** The measured nitrite (and nitrate) concentration and isotopic signatures in surface soil
collected in the Rongbuk Valley.

| Soil sample ID | $NO_2^-$ | | | | $NO_3^-$ | | | |
|---|---|---|---|---|---|---|---|---|
| | Conc.(ng g⁻¹) | $\delta^{15}N$ | $\delta^{18}O$ | $\Delta^{17}O$ | Conc.(ng g⁻¹) | $\delta^{15}N$ | $\delta^{18}O$ | $\Delta^{17}O$ |
| East-1 | 67.7 | -12.0 | 6.1 | 3.3 | 1127.3 | -2.7 | 23.4 | 5.7 |
| East-2 | 76.1 | -11.9 | 2.7 | 2.2 | 1098.3 | 1.3 | 18.1 | 3.8 |
| East-3 | 82.3 | -14.6 | 6.0 | 3.3 | 2978.1 | -0.3 | 25.4 | 6.6 |
| West-1 | 88.6 | -10.1 | 8.9 | 1.6 | 3176.5 | -3.1 | 44.5 | 13.4 |
| West-2 | 106.2 | -7.0 | 11.4 | 1.5 | 2880.3 | -1.2 | 22.1 | 6.2 |
| West-3 | 179.3 | -5.2 | 12.7 | 1.4 | 7686.9 | -3.4 | 35.8 | 11.4 |
| South-1 | 53.8 | -13.2 | 12.7 | 6.7 | 3651.2 | -0.6 | 32.6 | 10.1 |
| South-2 | 42.3 | -9.0 | 18.1 | 7.3 | 1683.2 | -2.2 | 49.3 | 14.8 |
| South-3 | 48.9 | -10.3 | 12.4 | 6.7 | 385.5 | -2.4 | 45.7 | 14.9 |


## 3.3 Surface soil nitrite concentration and isotopic signature

The concentration of soil $NO_2^-$ (and $NO_3^-$) as well as the corresponding isotopic signatures are
displayed in Table 1. The soil $NO_2^-$ content on the west slope of Rongbuk Valley (on average 124.7
ng g⁻¹) was higher than that observed on the east and south sides (75.3 ng g⁻¹ and 48.3 ng g⁻¹,
respectively). The mean surface soil $NO_2^-$ and $NO_3^-$ in the Rongbuk Valley were 82.8 and 2740.8 ng
g⁻¹, respectively. The soil $NO_3^-$ concentrations were significantly higher than the $NO_2^-$ by a factor of

8 ~ 40. In general, the measured soil $NO_3^-$ concentrations at the Rongbuk valley were significantly lower than other remote regions of TP (i.e., 23.1 µg g$^{-1}$ at Naqu, 8.4µg g$^{-1}$ at Yangbajing), while soil $NO_2^-$ concentrations were comparable to these reports (i.e., 90.3 ng g$^{-1}$ at Naqu,131.2 ng g$^{-1}$ at Yangbajing) (Wang et al., 2019). Soil $\delta^{15}N(NO_2^-)$ values ranged from -13.2‰ to -5.2‰ (on average -10.4‰), which are comparable to TSP $\delta^{15}N(NO_2^-)$ (-7.3‰). In comparison, we observed positive soil $\delta^{18}O(NO_2^-)$ and $\Delta^{17}O(NO_2^-)$, ranging from 2.7‰ to 18.1‰ (on average 10.5‰) and 1.4‰ to 7.3‰ (on average 3.8‰), respectively, in contrast to the negative $\delta^{18}O(NO_2^-)$ and near-zero $\Delta^{17}O(NO_2^-)$ observed in TSP samples. The determined soil $\delta^{18}O(NO_2^-)$ is comparable to that in laboratory incubated soil (11.8‰ ~ 12.5‰) (Lewicka-Szczebak et al., 2021).

## 4   Discussion

The significant contrast in $NO_2^-$ concentrations between TSP and PM$_{2.5}$ samples, as shown in Figure 1, suggests that at the sampling site atmospheric particle $NO_2^-$ overwhelmingly exists in coarse particles. This observation is consistent with previous studies across the TP, which also reported the absence of $NO_2^-$ in fine mode particles (PM$_{2.5}$ and PM$_{1.0}$) using either online real-time instrument or offline filter sampling (Decesari et al., 2010; Xu et al., 2020; Xu et al., 2023; Zhao et al., 2020), while relatively high levels of TSP $NO_2^-$ have been reported (Bhattarai et al., 2019; Bhattarai et al., 2023; Tripathee et al., 2017). In general, the chemical sources of particle $NO_2^-$ in the atmosphere encompass the uptake of HONO, particulate nitrate photolysis, and the $NO_2$-related reactions (i.e., photo-enhanced uptake of $NO_2$ on mineral dust, heterogeneous reaction of $NO_2$ on the surface of aerosol) (Nie et al., 2012; Vandenboer et al., 2014a; Chen et al., 2019; Shang et al., 2021), as summarized in Table 2. In addition to these in-situ atmospheric processes, growing evidence has revealed that long-range transport of atmospheric pollutants from South Asia also contributes considerably to aerosol loadings in TP in the spring (Kang et al., 2019; Bhattarai et al., 2023; Zhao et al., 2020), which may also bring nitrite along with other pollutants. Moreover, the lifting of surface dust can also contribute to the atmospheric coarse particles and significantly influence the chemical composition of TSP (Zhang et al., 2021a; Pokharel et al., 2019), and therefore soil nitrite could also be a potential source for TSP $NO_2^-$. In the following discussion, we examine the potential importance of the abovementioned processes to the observed high $NO_2^-$ content in coarse particle and discern the most likely ones.

**Table 2.** Particulate nitrite concentration and formation pathways/sources compiled in the literature.

| Site | Period | $NO_2^-$ Conc. (mean, ng m$^{-3}$) | Formation pathways/Sources | Reference |
|---|---|---|---|---|
| QOMS station | April 2017 | 60 | Biomass burning emission transported from South Asia | Bhattarai et al., 2023 |
| Bakersfield, California | May–July 2010 | 150 | HONO uptake on lofted alkaline soil particles. | VandenBoer et al., 2014 |
| Jinan, China | November 2013 − January 2014 | 2080 | Heterogeneous reactions of $NO_2$ | Wang et al., 2015 |
| Seoul, Korea | May−July, 2005 | 1410 | Heterogeneous reactions of $NO_2$ | Song et al., 2009 |
| Shanghai, China | June 2020 | 210 | Heterogeneous reactions of $NO_2$, reduction of $NO_2$ by S(IV) | Shang et al., 2021 |
| Mt.Heng, China | April 2009 | 2500 | Surface $TiO_2$ photocatalysis of $NO_2$ | Nie et al., 2012 |

**4.1 The potential effects of atmospheric chemistry on $NO_2^-$ in TSP**

386        Increasing evidence supports particulate nitrate photolysis as an important source of

atmospheric HONO especially in pristine atmosphere, with $NO_2^-$ serving as the intermediate in the
subsequent gas-particle partition process (Andersen et al., 2023; Ye et al., 2016). In theory,
particulate $NO_2^-$ (and HONO) produced from particulate nitrate photolysis might be associated with
extremely negative $\delta^{15}N$ values, due to the significant nitrogen isotopic fractionation effect during
nitrate photolysis (Erbland et al., 2013; Frey et al., 2009). For example, $NO_2^-$ in water of hypersaline
ponds and soil of McMurdo Dry Valleys, Antarctica, produced from the $NO_3^-$ photolysis, were
characterized by significantly negative $\delta^{15}N$ values (< -80‰) (Peters et al., 2014). However,
$\delta^{15}N(NO_2^-)$ in this study was only ~2‰ lower than the $\delta^{15}N(NO_3^-)$ in TSP samples collected during
this campaign (on average (-5.3 ± 3.3)‰, Text S1) and that at the QOMS stations (annual average
of (-5.1 ± 2.3) ‰) (Wang et al., 2020). The similarity in $\delta^{15}N$ isotopes between $NO_2^-$ and $NO_3^-$
suggests particulate nitrate photolysis is unlikely to be the primary source of the TSP $NO_2^-$.

398        In addition to nitrate photolysis, the absorption of HONO on alkaline aerosols (i.e., lofted dust

and road salt particles) can also result in accumulation of $NO_2^-$ into the particle phase (Vandenboer
et al., 2014a; Chen et al., 2019). For example, *VandenBoer et al.* observed a synchronous
enhancement of fine particle $NO_2^-$ (as high as 730 ng m$^{-3}$) alongside the buildup of HONO (up to
1.37 ppbv) after sunset in an agricultural site (Vandenboer et al., 2014a). However, the levels of
HONO in terrestrial background environments, typically on the order of dozens of pptv (Ye et al.,

2023), are obviously too low to support the observed unexpectedly high levels of particulate $NO_2^-$ (up to 1300 ng m$^{-3}$). What is more, previous studies indicted the HONO uptake predominately occurs on fine particles (Wang et al., 2015; Chen et al., 2019; Shang et al., 2021), while our observations indicated $NO_2^-$ only exists in coarse particles.

The uptake of $NO_2$ on mineral dust has also been identified as a significant route for the formation of particulate $NO_2^-$ or gas-phase HONO (Nie et al., 2012; Ndour et al., 2008). For example, *Nie et al.* found a significantly enhanced $NO_2^-$ in coarse particle during daytime in a dust storm event in Mt. Heng (up to 4.5 μg m$^{-3}$) (Nie et al., 2012). The proposed mechanism is initiated by photo-enhanced conversion of $NO_2$ to $NO_2^-$ via photo-produced electrons on surface of dust. Nevertheless, given the relatively small $NO_2$ uptake coefficients on mineral dust or salt surface (generally lower than $1\times10^{-6}$) (Yu et al., 2021; Bao et al., 2022; Xuan et al., 2025) and the low concentration of $NO_2$ in pristine environment of TP (e.g., ~ 140 pptv at Namco (Wang et al., 2023), and would be even lower at Mt. Qomolangma), such high levels of particulate $NO_2^-$ are beyond the capacity of $NO_2$ heterogeneous reactions.

Other than the above-mentioned rationales, the atmospheric physicochemical processes leading to $NO_2^-$ production would influence both fine and coarse-mode particles, and some of the processes (e.g., HONO uptake) preferentially interact with fine-mode particles. However, the observation indicated $NO_2^-$ only exists in TSP but not $PM_{2.5}$, suggesting atmospheric physicochemical processes are unlikely to account for the elevated levels of $NO_2^-$ in TSP.

**4.2 Potential effect of atmospheric pollutants in South Asia via long-range transport**

There is a growing body of compelling evidence indicating that the elevated aerosol loadings and chemical species in TP during spring, i.e., black carbon (Cong et al., 2015; Kang et al., 2019) and soluble components (Dasari et al., 2023; Bhattarai et al., 2023; Lin et al., 2021; Wang et al., 2020; Zhao et al., 2020) are significantly linked to biomass burning emissions from South Asia, which can penetrate into TP via long range transport. Recently, *Bhattarai et al.* observed synchronously elevated water-soluble nitrogen compounds (i.e., $NO_2^-$, $NH_4^+$, $NO_3^-$), levoglucosan (a molecule marker for biomass burning) and bulk $\delta^{15}N$ signatures in TSP at QOMS station, once upon the arrival of biomass burning plumes from South Asia (Bhattarai et al., 2023). Specially, they

found that TSP $NO_2^-$ averaged 60 ng m$^{-3}$ during the spring biomass burning influenced episodes,
while $NO_2^-$ was always below the detection limit in other seasons (Bhattarai et al., 2023). This
suggests that biomass burning events in South Asia and the subsequent long-range transport could
contribute to accumulation of $NO_2^-$ in aerosols in Tibet.
In this study, the mass concentrations and compositions of water-soluble ions, including $SO_4^{2-}$,
$NO_3^-$, $NH_4^+$ and TSP $NO_2^-$, also varied substantially throughout the springtime campaign (Figure 1).
The potential effect of South Asian pollutants on our observations was explored by analyzing the
air masses origins during the sampling period. As shown in Figure 5, during the first-half of the
campaign (i.e., before April 30$^{th}$), air masses predominately originated from or passed through
northern India and Nepal with intensive human activities and numerous fire hotspots (represented
by the dense red dots in Figure 5), indicating the potential impact of South Asia pollutants on aerosol
loadings of TP. Correspondingly, elevated concentrations of secondary inorganic ions (i.e., $NH_4^+$,
$NO_3^-$ and $SO_4^{2-}$) in TSP and PM$_{2.5}$ were observed before April 30$^{th}$, which are comparable to the
values of previous reports in QOMS station when arrived air masses experiencing severe biomass
burning emissions (Bhattarai et al., 2023; Lin et al., 2021). Meanwhile, elevated levels of TSP $NO_2^-$
(625 ± 457 ng m$^{-3}$) was observed during this biomass burning-impacted period, and $\delta^{15}N(NO_2^-)$
values exhibited substantial variability. TSP samples collected during daytime of April 27$^{th}$ and night
of April 28$^{th}$ were associated with high $NO_2^-$ concentrations and $\delta^{15}N(NO_2^-)$ values. In comparison,
TSP $NO_2^-$ in other samples were significantly $^{15}$N-depleted. Assuming that biomass burning
emission accounted for the two high $\delta^{15}N(NO_2^-)$ samples (Bhattarai et al., 2023), the observed
relatively low $\delta^{15}N(NO_2^-)$ in other samples before April 30$^{th}$ likely indicated the potential
contribution from additional emission sources.
From May 1$^{st}$ to May 6$^{th}$, air masses originated primarily from the inside of the TP or
surroundings and none of the fire hotspot was detected at the whole TP and along the air mass
trajectories (Figure 5), potentially excluding the influence of biomass burning and anthropogenic
emissions during this period. Accompanied by this significant shift in air mass origins, the
concentrations of $NH_4^+$, $NO_3^-$ and $SO_4^{2-}$ in both PM$_{2.5}$ and TSP, as well as the TSP $NO_2^-$ apparently
decreased. In particular, $NH_4^+$ in most PM$_{2.5}$ and TSP samples were below the detection limit during
this period. Furthermore, $K^+$ in PM$_{2.5}$, a common tracer for biomass burning, declined from 269 ±
432 ng m$^{-3}$ before April 30[th] to $22 \pm 12$ ng m$^{-3}$ after April 30[th], with the difference being statistically
significant ($p < 0.05$). Meanwhile, the average TSP $NO_2^-$ also declined significantly from $625 \pm 457$
ng m$^{-3}$ before April 30[th] to $147 \pm 145$ ng m$^{-3}$ after April 30[th]. Note substantially high levels of TSP
$NO_2^-$ were also observed on several days after April 30[th], i.e., during daytime of May 3[rd] ($\sim 400$ ng
m$^{-3}$). Although the TSP $NO_2^-$ concentrations varied in wide range after April 30[th], the $\delta^{15}N(NO_2^-)$
was relatively stable and comparable to that determined before April 30[th] (except for the two high
$\delta^{15}N(NO_2^-)$ samples).

468       In short summary, these results likely suggest the significant impact of South Asia pollutants

(i.e., biomass burning emissions) through long-range transport on the TSP $NO_2^-$, especially for
samples collected before April 30[th]. However, it is difficult for the long-range transport to explain
why $NO_2^-$ is predominately present in coarse particles, as fine mode particle is typically easier to be
transported in principle. While further studies need to be conducted to find out the exact reasons,
one possibility is that the size partition of fine particle $NO_2^-$ toward the coarse mode range during
transport. Similar size shifts of $NO_3^-$ have been observed and used to explain the enrichment of $NO_3^-$
in coarse-mode aerosols in marine environment (Matsumoto et al., 2009). In addition, the coarse
mode particles in this study contain more alkaline species (Figure 2), which makes nitrite more
stable in TSP during the transport or more stably exist in TSP after being uptake when the polluted
air masses reaching TP. For the period after April 30[th], since air masses originated from clean regions
with little to no biomass burning sources, other sources of nitrite might be required.

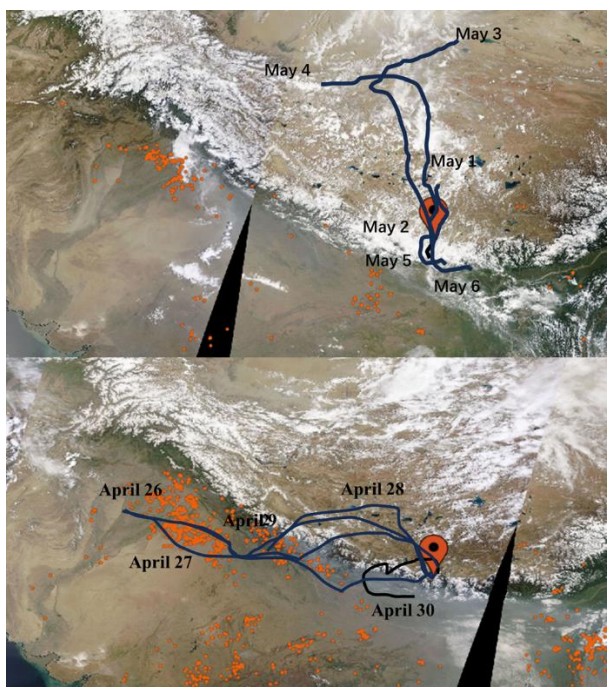


**Figure 5.** The modelled air-mass back trajectories during "Earth Summit Mission" scientific

expedition in spring 2022 (a: from May 1st to May 6th; b: April 26th to April 30th). The active fire

spots captured by MODIS (https://worldview.earthdata.nasa.gov, the red dots in the figure) are also

presented.

**4.3 The potential effects of lofted dust**

Previous reports have confirmed that wind-blown mineral dust contributed significantly to

coarse mode aerosols in TP (Kang et al., 2016; Zhang et al., 2021a). Notably, surface soil collected

from the Rongbuk valley is characterized with elevated $NO_2^-$ concentration (up to 180 ng g$^{-1}$, Table

1). The surface soil $NO_2^-$ is expected to mainly reside in the coarse mode after suspended in the

atmosphere (Drakaki et al., 2022), consistent with our observations that particle $NO_2^-$ was

predominately confined into coarse particle. The unique environment of Rongbuk valley,

characterized by exposed surface soil and strong wind (reaching as high as 9 m/s during this

campaign) could facilitate the resuspension of soil components into the atmosphere. Furthermore,

small localized tornadoes were frequently observed before April 30th, while the snow events

occurred in April 30th would reduce the soil-derived dust emission by increasing the snow coverage

and enriching the soil moisture. In fact, concentrations of $Ca^{2+}$ in TSP, which predominately

originate from dust emission, also declined to some extent after April 30th.

498       In addition, the similarity in $\delta^{15}N$ of $NO_2^-$ between TSP (-7.3 ± 3.1‰) and the surface soil (-10.3

± 3.0‰) also likely supports that locally emitted surface soil may contribute to the observed high
levels of TSP $NO_2^-$. But one should note that the oxygen isotopes ($\delta^{18}O$ and $\Delta^{17}O$) of TSP $NO_2^-$ were
significantly lower compared to that in soil $NO_2^-$, indicating that the original soil $NO_2^-$ oxygen
isotope may have been modified after resuspension. This discrepancy could be explained by the
potential oxygen isotope exchanges between TSP $NO_2^-$ and aerosol liquid water (fractionation effect
of $^{18}\varepsilon_{eq} \approx 16$‰ at local temperature, T = 270K, $^{18}\varepsilon_{eq} = -0.12$ T + 48.79 (Buchwald and Casciotti,
2013)), which tend to deplete both the initial $\delta^{18}O(NO_2^-)$ and $\Delta^{17}O(NO_2^-)$.
We noted the oxygen isotope exchange process between $NO_2^-$ and $H_2O$ also occurs in surface
soil. Previous study indicated that in high-altitude arid regions of TP (i.e., >5000m), denitrification
process dominated the surface soil $NO_2^-$ production, accounting for ~75% (Wang et al., 2019). Soil
$NO_2^-$ generated from denitrification process is expect to inherit the $\Delta^{17}O$ signatures of substrate $NO_3^-$.
In this study, the surface soil $\Delta^{17}O(NO_3^-)$ were positive with average values of 9.6‰ (Table 1). The
positive soil $\Delta^{17}O(NO_3^-)$ have been observed on arid environments (Wang et al., 2016), such as
desert soil, where the low water moisture favors the preservation of atmospherically derived $NO_3^-$.
One could estimate that soil $\Delta^{17}O(NO_2^-)$ derived from the denitrification would be 9.6‰, nitrite
from other sources (e.g., nitrification) should possess zero $\Delta^{17}O$, thus in total nitrite in soil should
possess $\Delta^{17}O$ of ~7.2‰ (estimated as $0.75 \times \Delta^{17}O(NO_3^-)$). However, the determined soil $\Delta^{17}O(NO_2^-)$
(3.8‰) is significantly lower compared to the estimated soil $\Delta^{17}O(NO_2^-)$, indicating the occurrence
of exchange process between soil water and $NO_2^-$, which would reduce the soil $\Delta^{17}O(NO_2^-)$ to some
extent. The exchange process between soil water and $NO_2^-$ is particularly evident in west slope of
Rongbuk Valley, where soil $\Delta^{17}O(NO_2^-)$ is as low as 1.5‰ while soil $\Delta^{17}O(NO_3^-)$ is on average
10.3‰. Soil $\Delta^{17}O(NO_2^-)$ should be erased to near-zero if exchange process between soil water and
$NO_2^-$ was efficient. Therefore, the fact that the observed soil $\Delta^{17}O(NO_2^-)$ remain above 0‰ indicates
unfavorable conditions for the oxygen isotope exchange process, likely due to the extremely low
soil moisture content (~1%) in surface soil (Ma et al., 2023).
Upon resuspension into atmosphere, the soil-derived dust aerosols usually exhibited a certain
degree of hygroscopicity (Tang et al., 2016; Chen et al., 2020), allowing the absorption of water
molecule on dust aerosol. For example, the aerosol water was determined to account for ~20% of

the total $PM_{10}$ mass during Saharan dust plumes (Cardoso et al., 2018). Based on laboratory experiment, Tang et al., 2019 reported that Asian dust also exhibited substantial hygroscopic property and revealed that the water-soluble inorganic ions, such as $Cl^-$, $SO_4^{2-}$ and $NO_3^-$ played a critical role in the absorption of water molecules on dust aerosol (Tang et al., 2019). In the present study, the $SO_4^{2-}$ and $NO_3^-$ account for ~30% of the total mass of water-soluble ions in TSP, implying the potential uptake of water vapor on aerosol surface. In addition to water-soluble ions, the hygroscopicity of mineral dust also depends on the surface areas, and wind-blown dust experiences a substantial increase in surface area after being lifted into the atmosphere, enhancing its capacity for water uptake (Chen et al., 2020; Seisel et al., 2004). The hygroscopicity of dust aerosol is expected to accelerate the oxygen isotope exchanges between $NO_2^-$ and aerosol liquid water. The atmospheric water vapor $\delta^{18}O$ isotope in TP is determined to be significantly negative (approximately -35‰ to -15‰ at a remote site in TP with altitude of ~ 4200m) (Yu et al., 2015). Similarly, the oxygen isotope exchange between $NO_2^-$ and $H_2O$ would also homogenize and erase original soil $\Delta^{17}O(NO_2^-)$ signals, because aerosol liquid water is characterized by negligible $\Delta^{17}O$ values (Luz and Barkan, 2005). Consequently, isotope exchange with aerosol water would further reduce both $\delta^{18}O$ and $\Delta^{17}O$ of TSP $NO_2^-$, effectively masking the original isotopic signature inherited from surface soil.

We also noted that $NO_2^-$ in surface soil is significantly lower than $NO_3^-$, by on average 35 times (Table 1), contrasting with the chemical compositions in TSP. It is important to note that during the complex dust generation and aerosolization processes, water-soluble ions exhibit significant chemical enrichment relative to that in parent soil (Wu et al., 2022; Gao et al., 2023). For example, Wu et al., 2022 reported that contents of nitrate in sand dust-derived $PM_{10}$ is higher than the original soil samples by 2~80 times, while sulphate can be up to 500 times higher than the original soil. Although the enrichment of nitrite has not been evaluated to the extent of our knowledge, we propose that the observed discrepancy of nitrate/nitrite ratio between TSP and surface soil can be reconciled if the nitrite is enriched more efficiently than nitrate during dust aerosolization, i.e., by a factor of 30. To further assess the potential contribution of resuspended soil to elevated nitrite in TSP, we conducted a rough estimation based on laboratory investigations from Wu et al. (2022). First, the springtime TSP concentrations observed in the nearby QOMS station is used ($65.1 \pm 50.9$

μg m$^{-3}$; (Liu et al., 2017)), as TSP mass concentrations surrounding the Base camp were not available. Second, we assume the nitrite content in soil-derived coarse particles to be similar to that of nitrate (0.2%) in laboratory-generated dust aerosol from natural sandy and gravel soils (Wu et al., 2022), which is comparable to the soil texture of the Rongbuk Valley. This assumption can also explain the comparable amounts of nitrite and nitrate observed in TSP samples in this study. Results shown that the concentrations of nitrite in TSP can be approximately $130 \pm 102$ ng m$^{-3}$, on the same order with the observed nitrite concentration in TSP after April 30$^{th}$ ($147 \pm 145$ ng m$^{-3}$) but substantially lower than that before April 30$^{th}$ ($625 \pm 457$ ng m$^{-3}$). Therefore, we speculate that the resuspension of surface soil may account for the observed TSP nitrite after April 30$^{th}$, whereas the biomass burning and soil together co-contributed to the high TSP nitrite before April 30$^{th}$. This is also consistent with the shift of air mass origins, which clearly indicated that airmasses before April 30$^{th}$ is significantly impacted by the biomass burning emissions, and after April 30$^{th}$ airmasses primarily originated from clean regions. While we acknowledge that this simplistic estimation is subject to substantial uncertainty, it provides a first-order assessment supporting the hypothesis that wind-blown soil dust contributes to coarse-mode particulate nitrite. Previously, wind-blown mineral dust has been verified as a potential source for aerosol water-soluble ions (Engelbrecht et al., 2016; Wu et al., 2022; Wu et al., 2012) and dust aerosol is also recognized as one of the important aerosol types over TP (Pokharel et al., 2019).

## 5 Conclusion and Implications

Unexpectedly high levels of NO$_2^-$ associated with coarse-particle were observed in the pristine environment at Mt. Qomolangma in the spring, 2022. After examining the potential contributions of various NO$_2^-$ sources with assistance from air mass back-trajectory and isotope analyses, we propose that both soil-derived nitrite and long-range transport of pollutants from South Asia may contribute to coarse-particle NO$_2^-$ during spring at Mt. Qomolangma. This is also consistent with previous reports showing that dust and biomass burning emission through long-range transport from South Asia are the predominant contributors to the springtime aerosol loadings over TP (Zhao et al., 2020; Pokharel et al., 2019). The nitrite concentrations and isotopes further indicated that soil-derived nitrite likely serves as a baseline source of atmospheric NO$_2^-$, maintaining the background levels of TSP NO$_2^-$ at this pristine site, reflected by the relatively stable isotopes when soil-derived

nitrite predominated. In addition, air masses originating from South Asia would result in elevated levels of $NO_2^-$ observed before April 30th by bringing additional biomass burning and anthropogenic pollutants, as evidenced by the more varied isotopes before April 30th compared to after that day when air mass origins shifted from South Asia to central and north Tibet. However, the detailed mechanisms of nitrite enriched on the coarse particle remain unknown and need further explorations.

In the atmosphere, photolysis of particle nitrite can produce OH radical and NO, the latter is essential for the formation of atmospheric oxidants and secondary aerosols. Moreover, the elevated levels of particle $NO_2^-$ may serve as an important HONO source through the gas-to-particle partition process (Vandenboer et al., 2014a), and the thermodynamic equilibrium between particulate nitrite and HONO ([pN(III)]/[HONO] ratio) is primarily governed by the particle acidity and liquid water content (LWC) in theory (Fountoukis and Nenes, 2007; Vandenboer et al., 2014a; Chen et al., 2019). Based on the observed TSP $NO_2^-$ and estimated ratio of [pN(III)]/[HONO] (from 4.8 to 10.6, Text S2), we can estimate the potential level of atmospheric HONO if the partition ever occurs at this site (Vandenboer et al., 2014b), and result indicates HOHO would be at 8 ~ 15 pptv, on the same order with the observations at a central Tibetan site (~ 30 pptv at Namco (Wang et al., 2023)). Given that TSP concentrations usually reach maximum during spring over TP, i.e., $65 \pm 51$ μg m$^{-3}$ at the nearby QOMS station (Liu et al., 2017), our findings suggest that the coarse-particle may serve as a potential source of atmospheric HONO and $NO_x$ assuming the TSP are associated with nitrite. Although the coarse-particle tend to deposit rapidly within hours, their potential to influence local atmospheric chemistry remains important to some extent, particularly considering the frequent dust events in TP (loose arid/semiarid surface, sparse vegetation, and strong winds.(Long et al., 2025)) and the ubiquity of long-range transport of biomass burning emissions from South Asia during this season. The impact of the TSP nitrite on the budget of NOx, HONO and OH radicals especially in the background atmosphere could be investigated using regional or global atmospheric transport model, once the detailed mechanism regarding the sources and chemistry of TSP nitrite been elucidated. In summary, our results highlight the need for further investigation into the sources, partitioning, and chemical reactivity of aerosol-phase nitrite, particularly in the pristine Tibetan Plateau, where even small inputs of $NO_x$ or HONO can disproportionately affect oxidant budgets and reactive nitrogen cycling

614

**Acknowledgments** We thank Dr. Yunhong Zhou for the help in air mass back trajectory analysis. The authors thank all researchers involved in the "Earth Summit Mission-2022" scientific expedition from Peking University, Hefei Institutes of Physical Science Anhui Institute of Optics and Fine Mechanics of Chinese Academy of Sciences, Minzu University of China.

**Financial support** L.G. acknowledges financial support from the National Natural Science Foundation of China (Awards: 41822605 and 41871051), and the National Key R&D Program of China (2022YFC3700701), the Innovation Program for Quantum Science and Technology (2021ZD0303101). Z. Z acknowledges financial support from the National Natural Science Foundation of China (Awards: 42273001), Key Research and Development Plan Project of Anhui Province (2022l07020032). P. C acknowledges financial support from the National Natural Science Foundation of China (Awards: 42371151).

**Conflicts of interest**

The authors declare that they have no conflicts of interest.

**Autor contribution:**

L.G designed the research, interpreted the data; L.G and Z.Z prepared the manuscript with contributions from all co-authors; Z.Z., Y.W., C.Y., T.Z., C.Z., Z.J., and P.C., conducted the field sampling and laboratory measurements; L.G, Z.Z., and P.C., acquired funding; L.G., S.K., and C.Y. reviewed and edited the manuscript. All authors have given approval to the final version of the manuscript.

**Data availability**

The data supporting the findings of this study are available in the archival repository at: https://doi.org/10.6084/m9.figshare.28188320.v1

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
