# Peer review of "Qomolangma 2 Zhongyi Zhang1, 2, Chunxiang Ye3, Yichao Wu1, Tao Zhou4, Pengfei Chen5,6, Shichang 3 Kang5,6, Chong Zhang3, Zhuang Jiang1, Lei Geng1,2,4\* 4 5 1Deep Space Exploration Laboratory/School of Earth and Space Sciences, University of Science 6 and Technology of China, Hefei 230026, Anhui, China 7 2CAS Center for Excellence in Comparative Planetology, University of Science and Technology of "

_EGUsphere, 2024_

## Referee Comment (RC2)

This study investigates the presence of high nitrite ($NO_2^-$) levels in coarse atmospheric particles at Mt. Qomolangma (Mount Everest) during a spring field campaign in 2022. The researchers found significant enrichment of $NO_2^-$ in total suspended particulates (TSP) but not in fine particles ($PM_{2.5}$). The study suggests that wind-blown soil, which contains high levels of $NO_2^-$, is likely the primary source of this enrichment. Additionally, long-range transport of pollutants from South Asia may contribute to elevated $NO_2^-$ levels, although the specific mechanisms remain unclear. The findings highlight the previously overlooked role of soil-derived $NO_2^-$ in atmospheric chemistry and its potential impact on the atmospheric oxidation capacity in remote regions like the Tibetan Plateau. However, it will be an even stronger paper if the following points are carefully considered. This manuscript can be published after minor revision.

Major comments:

1. The study highlights the potential for soil-derived $NO_2^-$ to influence atmospheric oxidation capacity through processes like photolysis or gas-particle partitioning. However, the broader implications for regional and global atmospheric chemistry are not fully explored. A more comprehensive discussion on how these findings fit into larger atmospheric models and their potential impact on climate and air quality would strengthen the study.

2. Although the manuscript provides a detailed description of the experimental procedures and results, the discussion on the research background and significance is not sufficiently in-depth. For example, while the importance of the Mt. Qomolangma region is mentioned, key questions such as why the study of $NO_2^-$ in coarse particles is important and the implications of this finding for the global atmospheric chemistry cycle are not thoroughly discussed.

Minor comments:

1. Line 23. "Atmospheric reactive nitrogen cycling, with nitrous acid (HONO) and particulate nitrite ($NO_2^-$) as important intermediates, is crucial for maintaining the atmospheric oxidation capacity of the background atmosphere on the Tibetan Plateau." Would be better.

2. Line 65, line 128. The terms "TSP" and "total suspended particulates" are used interchangeably. It would be beneficial to use one term consistently throughout the manuscript to avoid confusion.

3. Line 55. In Section "Introduction", the phrase "Atmospheric oxidation capacity (AOC) regulates the formation of secondary aerosol and the removal of trace gases

including $CH_4$" could be shortened to "Atmospheric oxidation capacity (AOC) regulates secondary aerosol formation and trace gas removal, including $CH_4$."

---

## Author Comment (AC1)

**Response to the referee #1**

*Understanding the chemistry of reactive nitrogen in pristine environments, such as the Tibetan Plateau, is crucial for advancing our knowledge of atmospheric chemistry. This study undertook a challenging field campaign, collecting $PM_{2.5}$, total particle, and soil samples to investigate the unusual enrichment of nitrite in coarse particles and its potential sources. While the effort in sample collection and extensive analysis is commendable, the presentation of the data in this paper lacks clarity, and the interpretation of the results is, in my opinion, not entirely sound.*

*After reviewing this manuscript, I remain unconvinced by the authors' arguments, as significant gaps in explanation persist. Given the high publication standards of Atmospheric Chemistry and Physics (ACP), I believe the current manuscript does not meet the criteria for publication without substantial revisions.*

**Response:** We thank the reviewer for your time and thoughtful evaluation of our manuscript. We appreciate the acknowledgment of the efforts involved in our field campaign and sample analysis. We also recognized and took seriously the concerns raised regarding the clarity of data presentation and the interpretation of our results. But before going with details of response, we wanted to first clarify that probably due to the organization and writing in the original manuscript, our intentions may have not been clearly conveyed. In this manuscript, the objective was not to draw a definitive conclusion on the source/origin of particulate nitrite, instead we wanted to present this finding and discuss what are the potential sources, as reflected by the title "**On the presence of high nitrite ($NO_2^-$) in coarse particles at Mt. Qomolangma**". We thought the finding, i.e., the presence of nitrite exclusively in the coarse particle is interesting and puzzling, and tried to explore the possible sources/origins and implications of this finding. We realized that it is difficult to discern the exact origin/source of the observed coarse particle nitrite given current dataset available, we hypothesized that long-range transport of pollutants from South Asia and wind-blown soil together may explain the observations, though additional investigations are necessary to discern the exact mechanisms how these sources contribute to nitrite in TSP, or if there are other sources. We are making these points clearer in the revised manuscript. Please review our point-by-point responses to your comments with more details in the following point-to-point responses.

*Here are my major concerns:*

*1. Uncertainties in ion concentrations of $PM_{2.5}$ and total particles. The comparison of ion concentrations between TSP and $PM_{2.5}$ is particularly interesting. It clearly shows that nitrite ions are present only in TSP, while $PM_{2.5}$ contains none. However, what stands out is that despite $PM_{2.5}$ being a subset of TSP, some $PM_{2.5}$ samples occasionally show higher ion concentrations (in µg/m³) than their corresponding TSP samples. For instance, the April 30 $PM_{2.5}$ sample appears to contain more nitrate than the TSP sample.*

*I suspect this discrepancy may stem from analytical uncertainties in ion concentration measurements or uncertainties in the blank corrections, but this issue has not been addressed in the manuscript. Understanding these uncertainties is crucial, especially since the ion composition of coarse aerosols is determined by subtracting two similar measurements.*

*I also suggest that the authors directly present the ion concentrations of coarse aerosols. While this may sometimes result in negative values, it would provide a clearer picture of the uncertainties associated with ion concentration measurements.*

**Response:** Thanks for the valuable and insightful suggestions. Indeed, there might be some degree of uncertainties given the measurements themselves and conversion of liquid concentrations to air mass concentrations. The latter are subject to further uncertainties given the parameters of the sampling instruments (e.g., sampling flow rate). In the revised manuscript, we have added the uncertainties regarding the ionic concentrations in $PM_{2.5}$ and TSP, as outlined in Section 2.3 (lines 158-188):

**2.3 Ionic concentration analysis and uncertainty estimation**

Water-soluble inorganic ions (WSIs, including $Na^+$, $NH_4^+$, $K^+$, $Mg^{2+}$, $Ca^{2+}$, $Cl^-$, $NO_2^-$, $NO_3^-$, and $SO_4^{2-}$) in TSP (entire filter, ~16.6 cm²) and $PM_{2.5}$ (1/32 section, ~13.0 cm²) were extracted using 20 mL of Milli-Q ultrapure water (18.2Ω cm) in an ultrasonic bath at room temperature for 30 min. Note the TSP filters were cut to fit the filter holder from the standard Whatman quartz-fiber filters which were also used as the $PM_{2.5}$ filters. After filtration through a 0.22 μm pore size syringe filter which was pre-cleaned with ultrapure water, the filtrate was subjected to inorganic species analysis using ion chromatography (Dionex Aquion) (Zhang et al., 2020). Blank filters were pretreated and measured the same as real samples, and the limits of detection (LOD) was calculated as 3 times of standard deviations of blanks (Fang et al., 2015). In general, $Na^+$ in the blank filters is comparable to samples, a well-known issue for the Whatman quartz filter which is high in $Na^+$ blank. Therefore, in this study, we discarded the $Na^+$ data. The volatile components (i.e., $NH_4^+$, $NO_2^-$, $NO_3^-$) and $K^+$ in blank filters are low but several times higher than the detection limits; $SO_4^{2-}$, $Mg^{2+}$ and $Ca^{2+}$ in blank are significantly higher than the respective LOD but lower than samples by at least five times (the lower end). All reported concentrations of each ion were blank corrected as follows:

$$C_i = \frac{(\rho_{sample} - \rho_{blank}) \times V_{water} \times F}{V_{air}}$$

with $C_i$ representing the ambient concentrations of specie $i$ in air (ng m⁻³ or μg m⁻³), $\rho_{sample}$ and $\rho_{blank}$ are the concentrations determined by the ion chromatography (ng mL⁻¹), $V_{water}$ is the volume of ultrapure water used for extraction (20mL), $V_{air}$ is the volume of air sampled for each $PM_{2.5}$ or TSP filter, F is the ratio of particulate matter collection area for $PM_{2.5}$ or TSP filters to the filter area used for extraction.

The overall uncertainty in ion concentration was estimated according to the law of error propagation, accounting for the sampling air volume (3% for $PM_{2.5}$ samples and 1% for TSP samples as provided by the manufactures), the extraction of water volume (~0.3% for pipetting from the manufacture Eppendorf), the blanks, and the analytical uncertainty from ion chromatography and calibration, assuming that these factors are independent. The analytical uncertainty for water-soluble ions concentration determination using ionic chromatography has been extensively assessed in our laboratory, with values typically <5% for all inorganic species at concentration of 500 ng mL⁻¹. The combined uncertainty about the ionic concentrations in $PM_{2.5}$ and TSP are shown in Table S1. In general, TSP samples are associated with relatively high overall uncertainty compared to $PM_{2.5}$ samples, perhaps due to the relatively high blank variability due to the low mass loading in TSP.

In the revised manuscript, as suggested, we have also presented the ion compositions in the coarse-mode particulate in new Figure 2. We acknowledge that in some cases, the ions concentrations in $PM_{2.5}$ is higher than that in TSP filters, as indicated by the discontinuous time series of certain ions,

i.e., $SO_4^{2-}$. Nevertheless, this abnormal phenomenon has no significant impact on the key findings and interpretations in the manuscript. For example, the predominance of nitrite in coarse particulate matter remains unambiguous and significant declines of secondary inorganic ions after April 30th in response to the shift of air mass origins remains well-supported.

[Figure]

**Figure 2.** The estimated chemical compositions and time series of mass concentrations of chemical species ($NO_2^-$, $SO_4^{2-}$, $NO_3^-$, $Ca^{2+}$, etc.) in coarse-mode particulate during the springtime campaign (calculated as the differences between TSP and $PM_{2.5}$). Discontinuities in the time series were observed for certain species (e.g., $SO_4^{2-}$), likely resulting from relatively lower concentrations in the TSP samples compared to those in the corresponding $PM_{2.5}$ samples. This discrepancy is likely attributed to the propagated uncertainties involved in the concentration analysis, sampling approach and blank corrections.

*2. Missing evidence: mass balance in coarse particles. I would like to see a little bit more discussion in section 4.3 when the authors attempted to attribute the observed nitrite to lofted dust, maybe as simple as mass balance calculations. For example, if the observed nitrite is indeed coming from soil, to get ~1 ug/m³ of nitrite from soil, how much soil do you need in the air giving average soil nitrite concentration of < 100 ng/g? About 10 g/ m³. Then, does the observed TSP concentration support your hypothesis?*

*Similarly, nitrate/nitrite ratio in soil also do not fully support authors' hypothesis, the authors argue that it is likely nitrate/nitrite distributed in particles of different sizes but there is no evidence supporting this, nor is there any previous work mentioned such effect. Therefore, I am not fully convinced by the existing evidence that soil is the main source of particle nitrite.*

**Response:** Thank you for your valuable suggestion. We have included a simple mass balance-based estimation in Section 4.3 (The potential effects of lofted dust) to further assess the potential contribution of resuspended surface soil to the observed nitrite ($NO_2^-$) concentrations in TSP. The result indicates that, under justified reasonable assumptions, soil nitrite can fully explain the TSP nitrite in the second period of the observations, i.e., after April 30th. For elevated levels of TSP

nitrite before the April 30th, biomass burning would be necessary to be included. This in fact reconciles our conclusion that biomass burning and soil together may explain the observations.

Below we listed our detailed assessments on the potential contribution of soil to TSP nitrite and our revisions inspired by the suggestion. The assessments are as follows (lines 541-570): "We also noted that $NO_2^-$ in surface soil is significantly lower than $NO_3^-$, by on average 35 times (Table 1), contrasting with the chemical compositions in TSP. It is important to note that during the complex dust generation and aerosolization processes, water-soluble ions exhibit significant chemical enrichment relative to that in parent soil (Wu et al., 2022; Gao et al., 2023). For example, Wu et al., 2022 reported that contents of nitrate in sand dust-derived $PM_{10}$ is higher than the original soil samples by 2~80 times, while sulphate can be up to 500 times higher than the original soil. Although the enrichment of nitrite has not been evaluated to the extent of our knowledge, we propose that the observed discrepancy of nitrate/nitrite ratio between TSP and surface soil can be reconciled if the nitrite is enriched more efficiently than nitrate during dust aerosolization, i.e., by a factor of 30.

To further assess the potential contribution of resuspended soil to elevated nitrite in TSP, we conducted a rough estimation based on laboratory investigations from Wu et al. (2022). First, the springtime TSP concentrations observed in the nearby QOMS station is used ($65.1 \pm 50.9$ μg m$^{-3}$; Liu et al., 2017), as TSP mass concentrations surrounding the Base camp were not available. Second, we assumed the nitrite content in soil-derived coarse particles to be similar to that of nitrate (0.2%) in laboratory-generated dust aerosol from natural sandy and gravel soils (Wu et al., 2022), which is comparable to the soil texture of the Rongbuk Valley. This assumption can also explain the comparable amounts of nitrite and nitrate observed in TSP samples in this study. Results shown that the concentrations of nitrite in TSP can be approximately $130 \pm 102$ ng m$^{-3}$, on the same order with the observed nitrite concentration in TSP after April 30th ($147 \pm 145$ ng m$^{-3}$) but substantially lower than that before April 30th ($625 \pm 457$ ng m$^{-3}$). Therefore, we speculate that the resuspension of surface soil may account for the observed TSP nitrite after April 30th, whereas the biomass burning and soil together co-contributed to the high TSP nitrite before April 30th. This is also consistent with the shift of air mass origins, which clearly indicated that airmasses before April 30th is significantly impacted by the biomass burning emissions, and after April 30th airmasses primarily originated from clean regions. While we acknowledge that this simplistic estimation is subject to substantial uncertainty, it provides a first-order assessment supporting the hypothesis that wind-blown soil dust contributes to coarse-mode particulate nitrite. Previously, wind-blown mineral dust has been verified as a potential source for aerosol water-soluble ions (Engelbrecht et al., 2016; Wu et al., 2022; Wu et al., 2012) and dust aerosol is also recognized as one of the important aerosol types over TP (Pokharel et al., 2019)."

The above assessments were based on several considerations. First, recent studies have indicated that due to the aerosol-soil fractionation during dust emission, the contents of water-soluble ions in soil-derived dust aerosol are significantly higher than the parent soil (Wu et al., 2022; Gao et al., 2023). Wu et al. (2022) conducted a detailed investigation of the chemical composition of water-soluble ions in soil-derived aerosols using a laboratory dust-generation system, which ensured consistency in particle size distribution and chemical composition between the generated particles and ambient dust. Their results serve as a reliable reference for estimating the potential contribution of specific components in soil-derived aerosols. Their results shown that the enrichment varied

significantly among the water-soluble ions. For example, contents of nitrate in sand dust-derived $PM_{10}$ is higher than the original soil samples by 2~80 times (on average 22 times), while sulphate can be up to 500 times higher than the original soil. Unfortunately, the enrichment of nitrite was not evaluated.

Second, in the present study, the ratio of nitrate to nitrite in surface soil is on average 30 (Table 1 in the main text), whereas the nitrate is comparable to the nitrite in the TSP samples. To reconcile this discrepancy, we propose that nitrite may experience more efficient enrichment than nitrate during aerosolization, i.e., nitrite was enriched ~30 times more than nitrate. Moreover, Wu et al. suggested that the nitrate contents in the sand dust-derived $PM_{10}$ is on average 0.2%. Since nitrate was determined to be comparable to the nitrite in the TSP samples, the content of nitrite in the soil-derived coarse particle is also assumed to be 0.2% in the our estimation. We acknowledge although the simplistic estimation is associated with significant uncertainty, it provides a way to assess the potential contribution of surface soil-derived nitrite to the observed high nitrite in TSP.

Third, the springtime TSP mass concentrations in the nearby QOMS stations ($65.1 \pm 50.9$ μg m$^{-3}$) were used, as there were no available reports regarding the TSP mass concentrations at the Base Camp. We estimated that the concentrations of nitrite in TSP can be approximately $130 \pm 102$ ng m$^{-3}$, on the same order with the observed nitrite concentration in TSP after April 30th but substantially lower than that before April 30th. Therefore, we suggest that resuspension of surface soil is a potential contributor to the TSP nitrite after April 30th. However, for the higher nitrite levels observed before April 30th, additional sources, such as biomass burning are likely required. This interpretation is supported by the corresponding shift in air mass origins before and after April 30th, with the earlier period showing significant influence from biomass burning emissions.

*3. Isotopic results do not seem to support authors' argument. The isotopic results also do not support the authors' argument. The $O^{17}$ signal in soil samples range from 1.4‰ to 7.3‰ but in TSP the $O^{17}$ is 0-1‰, a clear discrepancy. $\delta^{18}O$ also are significant different – 2‰ to 18‰ in the soil but lots of negative values in the TSP. I do not think this can be simply explained by isotope exchange with water because such isotope exchange always occurs: we need more evidence to believe that exchange never happens when the particles are on the surface, then once it was lifted into the air, within hours (typical lifetime of coarse particles) the exchange suddenly occurred.*

**Response:** Thanks for raising this important question. We agree that oxygen isotopic exchange between nitrite and water molecule occurs not only in the atmosphere but also in surface soils. The significant $\Delta^{17}O$ difference observed between nitrite and nitrate in surface soils (3.7‰ vs. 9.6‰) suggests that isotope exchange is indeed occurring in the soil environment as well. Moreover, the hygroscopic nature of dust aerosol enables the absorption of water and thus facilitate the exchange process between TSP nitrite and aerosol water. According to your suggestion, the discussion is revised as follows (lines 495-540):

[revised manuscript text omitted]

In brief, an extensive field investigation revealed active denitrification process in dryland surface soil at elevations above 5000m from Tibetan Plateau (similar to the soil texture around our sampling site), accounting for 75% of surface soil nitrite production (Wang et al., 2019). Surface soil nitrite

produced from denitrification process inherit $\Delta^{17}O$ of nitrate, since this process should follow the mass-dependent fractionation law. Based on the field investigations in Wang et al., 2019, we estimate the surface soil $NO_2^-$ should be associated with relatively higher $\Delta^{17}O$ values (~7.2‰, 9.6‰×0.75, assuming nitrite from other sources, e.g., nitrification, possess zero $\Delta^{17}O$) than our determined values (3.7‰). Therefore, the occurrence of oxygen atom exchange between nitrite and water molecule in surface soil may also occur, resulting in relatively low $\Delta^{17}O$ values in surface nitrite. The exchange process maybe particular evident in west slope soil, given the significant lower surface soil $\Delta^{17}O(NO_2^-)$ relative to $\Delta^{17}O(NO_3^-)$ (1.5‰ versus 10.3‰). Note that if the oxygen exchange process was sufficiently efficient, the surface soil $\Delta^{17}O(NO_2^-)$ should be erased to be negligible. Therefore, the determined positive surface soil $\Delta^{17}O(NO_2^-)$ probably indicated unfavorable conditions for the oxygen isotope exchange process may due to the extremely low soil moisture content (~1%) in surface soil (Ma et al., 2023).

Second, the mineral dust aerosols have been shown to exhibit hygroscopic properties, which would facilitate the oxygen exchange process between nitrite and water molecule, once emitted into atmosphere (Kumar et al., 2009; Shi et al., 2008; Tang et al., 2019). It is suggested that the water-soluble inorganic ions, especially sulfate, nitrate and ammonium majorly determine the hygroscopic properties of mineral dust aerosol (Tang et al., 2016; Shi et al., 2008; Tang et al., 2019), and these processes can result in a water content of up to 20% of the total $PM_{10}$ mass as observed in Saharan dust plumes (Cardoso et al., 2018). In the present study, the sulfate and nitrate account for ~30% of the total mass of water-soluble ions, implying the potential uptake of water vapor on aerosol surface. We suggested that the hygroscopic properties may facilitate the oxygen exchange between TSP $NO_2^-$ and water molecule, upon resuspension into atmosphere.

*4. More discussion needed for air mass from different regions. The back-trajectory analysis clearly shows that the field campaign sampled air from two distinct regions, with corresponding differences in ion concentrations and $d^{15}N$ values. However, the discussion of these differences is too simplistic. It would be beneficial for the authors to separately analyze how the aerosols from each period differ and to explore in greater depth how nitrite concentrations and isotopic compositions varied between them, as the differences are quite significant.*

*Additionally, while the authors suggest that long-range transport is unimportant for the nitrite budget, their argument in the final section contradicts this claim. To strengthen the manuscript, the discussion should remain consistent and logically cohesive.*

**Response:** Thank you for the valuable suggestion. We have expanded Section 4.2 (Potential effect of biomass burning emissions in South Asia via long-range transport) to separately analyze and compare the aerosol characteristics, nitrite concentrations, and isotopes before and after April 30[th]. We also clarified our interpretation of long-range transport of biomass burning emissions, emphasizing that while it likely plays a significant role in the observed high TSP $NO_2^-$ before April 30[th], its contribution to TSP $NO_2^-$ is minor after April 30[th]. The potential impact of anthropogenic and biomass burning pollutant via long-range transport was explored in-depth and the section 4.2 was modified to keep consistency according to your suggestions. In sum, we were not trying to state that biomass burning is not important, instead we acknowledge its contribution to TSP nitrite in the first half of the sampling campaign. We have made this clearer in the revision (lines 433-476).

[revised manuscript text omitted]

*5. Implications to AOC is weak. Since the source of nitrite remains unclear, the discussion on how lofted soil influences atmospheric chemistry is weak. The authors provide no data on atmospheric dust concentrations and do not address the transport potential of soil particles, which is likely limited due to their larger size. As a result, the final section lacks convincing supporting evidence.*

**Response:** Thank you for the valuable suggestion. We have revised this section to focus more on the potential atmospheric implications of elevated coarse-mode nitrite observed at Mt. Qomolangma, and the influence of lofted soil on atmospheric chemistry was reshaped accordingly. We now highlight its potential contribution to local photochemistry through HONO and NOx formation, supported by estimated HONO levels derived from observed [pN(III)]/[HONO] ratios. We also clarified that while these effects may be spatially limited due to the rapid deposition of coarse particles, they could still significantly impact local atmospheric oxidation capacity (AOC) in this pristine high-altitude environment, especially considering the ubiquity of the potential sources, i.e., the frequent biomass burning episodes through long range transport from South Asia (Bhattarai et al., 2023) and frequent dust activities in TP (Long et al., 2025).

Unexpectedly high levels of $NO_2^-$ associated with coarse-particle were observed in the pristine environment at Mt. Qomolangma in the spring, 2022. After examining the potential contributions of various $NO_2^-$ sources with assistance from air mass back-trajectory and isotope analyses, we suggest that both soil-derived nitrite and long-range transport of pollutants from South Asia may contribute to coarse-particle $NO_2^-$ during spring at Mt. Qomolangma. This is also consistent with previous reports showing that dust and biomass burning emission through long-range transport from South Asia are the predominant contributors to the springtime aerosol loadings over TP (Zhao et al., 2020; Pokharel et al., 2019). The nitrite concentrations and isotopes further indicated that soil-derived nitrite likely serves as a baseline source of atmospheric $NO_2^-$, maintaining the background levels of TSP $NO_2^-$ at this pristine site, reflected by the relatively stable isotopes when soil-derived nitrite predominated. In addition, air masses originating from South Asia would result in elevated levels of $NO_2^-$ observed before April 30th by bringing additional biomass burning and anthropogenic pollutants, as evidenced by the more varied isotopes before April 30th compared to after that day when air mass origins shifted from South Asia to central and north Tibet. However, the detailed mechanisms of nitrite enriched on the coarse particle remain unknown and need further explorations.

In the atmosphere, photolysis of particle nitrite can produce OH radical and NO, the latter is essential for the formation of atmospheric oxidants and secondary aerosols (Figure 6). Moreover, the elevated levels of particle $NO_2^-$ may serve as an important HONO source through the gas-to-particle partition process (Vandenboer et al., 2014a), and the thermodynamic equilibrium between particulate nitrite and HONO ([pN(III)]/[HONO] ratio) is primarily governed by the particle acidity and liquid water content (LWC) in theory (Fountoukis and Nenes, 2007; Vandenboer et al., 2014a; Chen et al., 2019). Based on the observed TSP $NO_2^-$ and estimated ratio of [pN(III)]/[HONO] (from 4.8 to 10.6, Text S2), we can estimate the potential level of atmospheric HONO if the partition ever occurs at this site (Vandenboer et al., 2014b), and result indicates HOHO would be at 8 ~ 15 pptv, on the same order with the observations in the background atmosphere at a central Tibetan site (i.e., ~ 30 pptv at Namco (Wang et al., 2023)). Given that TSP concentrations usually reach maximum during spring over TP, i.e., $65 \pm 51$ µg m$^{-3}$ at the nearby QOMS station (Liu et al., 2017), our findings suggest that the coarse-particle may serve as a potential source of atmospheric HONO and $NO_x$ assuming the TSP are associated with nitrite. Although the coarse-particle tend to deposit rapidly within hours,

their potential to influence local atmospheric chemistry remains important to some extent, particularly considering the frequent dust events in TP (loose arid/semiarid surface, sparse vegetation, and strong winds. Long et al., 2025) and the ubiquity of long-range transport of biomass burning emissions from South Asia during this season. The impact of the TSP nitrite on the budget of NOx, HONO and OH radicals especially in the background atmosphere could be investigated using regional or global atmospheric transport model, once the detailed mechanism regarding the sources and chemistry of TSP nitrite been elucidated. In summary, our results highlight the need for further investigation into the sources, partitioning, and chemical reactivity of aerosol-phase nitrite, particularly in the pristine Tibetan Plateau, where even small inputs of $NO_x$ or HONO can disproportionately affect oxidant budgets and reactive nitrogen cycling.

---

## Author Comment (AC2)

**Response to the referee #2**

*This study investigates the presence of high nitrite ($NO_2^-$) levels in coarse atmospheric particles at Mt. Qomolangma (Mount Everest) during a spring field campaign in 2022. The researchers found significant enrichment of $NO_2^-$ in total suspended particulates (TSP) but not in fine particles ($PM_{2.5}$). The study suggests that wind-blown soil, which contains high levels of $NO_2^-$, is likely the primary source of this enrichment. Additionally, long-range transport of pollutants from South Asia may contribute to elevated $NO_2^-$ levels, although the specific mechanisms remain unclear. The findings highlight the previously overlooked role of soil-derived $NO_2^-$ in atmospheric chemistry and its potential impact on the atmospheric oxidation capacity in remote regions like the Tibetan Plateau. However, it will be an even stronger paper if the following points are carefully considered. This manuscript can be published after minor revision.*

**Response:** We thank the referee for the thoughtful and concise summary of our work. We have revised the manuscript according to your insightful suggestion.

*Major comments: 1. The study highlights the potential for soil-derived $NO_2^-$ to influence atmospheric oxidation capacity through processes like photolysis or gas-particle partitioning. However, the broader implications for regional and global atmospheric chemistry are not fully explored. A more comprehensive discussion on how these findings fit into larger atmospheric models and their potential impact on climate and air quality would strengthen the study.*

Response: Thank you for this valuable suggestion. Although we can attribute the observed high levels of TSP nitrite to biomass burning and soil emission, the detailed mechanisms for the sources and atmospheric chemistry of TSP nitrite is still unknown and needs further exploration. The implication section has been revised according as follows (lines 587-610): "In the atmosphere, photolysis of particle nitrite can produce OH radical and NO, the latter is essential for the formation of atmospheric oxidants and secondary aerosols (Figure 6). Moreover, the elevated levels of particle $NO_2^-$ may serve as an important HONO source through the gas-to-particle partition process (Vandenboer et al., 2014a), and the thermodynamic equilibrium between particulate nitrite and HONO ([pN(III)]/[HONO] ratio) is primarily governed by the particle acidity and liquid water content (LWC) in theory (Fountoukis and Nenes, 2007; Vandenboer et al., 2014a; Chen et al., 2019). Based on the observed TSP $NO_2^-$ and estimated ratio of [pN(III)]/[HONO] (from 4.8 to 10.6, Text S2), we can estimate the potential level of atmospheric HONO if the partition ever occurs at this site (Vandenboer et al., 2014b), and result indicates HOHO would be at 8 ~ 15 pptv, on the same order with the observations in the background atmosphere at a central Tibetan site (i.e., ~ 30 pptv at Namco (Wang et al., 2023)). Given that TSP concentrations usually reach maximum during spring over TP, i.e., $65 \pm 51$ µg m$^{-3}$ at the nearby QOMS station (Liu et al., 2017), our findings suggest that the coarse-particle may serve as a potential source of atmospheric HONO and $NO_x$ assuming the TSP are associated with nitrite. Although the coarse-particle tend to deposit rapidly within hours, their potential to influence local atmospheric chemistry remains important to some extent, particularly considering the frequent dust events in TP (loose arid/semiarid surface, sparse vegetation, and strong winds. Long et al., 2025) and the ubiquity of long-range transport of biomass burning emissions from South Asia during this season. The impact of the TSP nitrite on the budget of NOx, HONO and OH radicals especially in the background atmosphere could be investigated using regional or global atmospheric transport model, once the detailed mechanism regarding the

sources and chemistry of TSP nitrite been elucidated. In summary, our results highlight the need for further investigation into the sources, partitioning, and chemical reactivity of aerosol-phase nitrite, particularly in the pristine Tibetan Plateau, where even small inputs of $NO_x$ or HONO can disproportionately affect oxidant budgets and reactive nitrogen cycling."

2. Although the manuscript provides a detailed description of the experimental procedures and results, the discussion on the research background and significance is not sufficiently in-depth. For example, while the importance of the Mt. Qomolangma region is mentioned, key questions such as why the study of $NO_2^-$ in coarse particles is important and the implications of this finding for the global atmospheric chemistry cycle are not thoroughly discussed.
**Response:** Thank you for this suggestion. We have added the research background and significance in various part in the revised manuscript.
In introduction section, we now added the following descriptions regarding on the importance of particle $NO_2^-$ as follows (lines 59-72): "Recent field campaign further highlighted the rapid reactive nitrogen cycling, with nitrous acid (HONO) and particulate nitrite ($NO_2^-$) as important intermediate, also plays an important role in maintaining the strong AOC in TP (Wang et al., 2023). For example, *Wang et al.* reported high-than-expected HONO (~30 ± 13 pptv) in the Namco station, a typical background site in the middle of TP, with HONO sources including $NO_2$ heterogenous conversion, soil emission and particulate nitrate photolysis (Wang et al., 2023). However, a detailed HONO budget analysis indicated these three dominant sources could not account for the observed daytime HONO levels at this background site, implying the existence of additional, yet unidentified, HONO sources. Particulate nitrite likely represents a potential source of HONO through thermodynamic partitioning processes, provided particulate nitrite may be present in significant amounts under favorable atmospheric conditions (Vandenboer et al., 2014a; Chen et al., 2019; Li, 1994). Interestingly, relatively high levels of nitrite ($NO_2^-$) in total suspended particulate (TSP) have also been reported from remote sites of TP, i.e., in a forest site in the Southeast Tibet (~ 140 ng m$^{-3}$) and at the Qomolangma monitoring station (QOMS, ~ 60 ng m$^{-3}$) (Bhattarai et al., 2019; Bhattarai et al., 2023)."
Regarding the implication of our findings, we have revised according to you last comment, please see our response to your last comment.

*Minor comments:*
*1. Line 23. "Atmospheric reactive nitrogen cycling, with nitrous acid (HONO) and particulate nitrite ($NO_2^-$) as important intermediates, is crucial for maintaining the atmospheric oxidation capacity of the background atmosphere on the Tibetan Plateau." Would be better.*
**Response:** Thank you. The abstract was revised accordingly.

2. Line 65, line 128. The terms "TSP" and "total suspended particulates" are used interchangeably. It would be beneficial to use one term consistently throughout the manuscript to avoid confusion.
**Response:** Thank you for pointing this out. We have revised the manuscript to use the abbreviation TSP (total suspended particulates)" consistently after its first definition to ensure clarity and avoid confusion

3. Line 55. In Section "Introduction", the phrase "Atmospheric oxidation capacity (AOC) regulates

the formation of secondary aerosol and the removal of trace gases including $CH_4$" could be shortened to "Atmospheric oxidation capacity (AOC) regulates secondary aerosol formation and trace gas removal, including $CH_4$.

**Response:** Thank you. The sentence was revised accordingly.

**Reference:**

Wang, J., Zhang, Y., Zhang, C., et al.: Validating HONO as an Intermediate Tracer of the External Cycling of Reactive Nitrogen in the Background Atmosphere, Environ Sci Technol, 57, 5474-5484, 10.1021/acs.est.2c06731, 2023.

Bhattarai H, Wu G, Zheng X, et al. Wildfire-derived nitrogen aerosols threaten the fragile ecosystem in Himalayas and Tibetan Plateau[J]. Environmental Science & Technology, 2023, 57(25): 9243-9251.

Bhattarai H, Zhang Y L, Pavuluri C M, et al. Nitrogen speciation and isotopic composition of aerosols collected at Himalayan forest (3326 m asl): seasonality, sources, and implications[J]. Environmental Science & Technology, 2019, 53(21): 12247-12256.

VandenBoer, T., Markovic, M., Sanders, J., et al. Evidence for a nitrous acid (HONO) reservoir at the ground surface in Bakersfield, CA, during CalNex 2010, Journal of Geophysical Research: Atmospheres, 119, 9093-9106, 2014a.

VandenBoer, T. C., Young, C. J., Talukdar, R. K., et al. Nocturnal loss and daytime source of nitrous acid through reactive uptake and displacement, Nature Geoscience, 8, 55-60, 10.1038/ngeo2298, 2014b.

Fountoukis C, Nenes A. ISORROPIA II: a computationally efficient thermodynamic equilibrium model for $K^+$–$Ca^{2+}$–$Mg^{2+}$–$NH_4^+$–$Na^+$–$SO_4^{2-}$–$NO_3^-$—$Cl^-$–$H_2O$ aerosols[J]. Atmospheric Chemistry and Physics, 2007, 7(17): 4639-4659.

Chen Q, Edebeli J, McNamara S M, et al. HONO, particulate nitrite, and snow nitrite at a midlatitude urban site during wintertime[J]. ACS Earth and Space Chemistry, 2019, 3(5): 811-822.

Long H, Cheng L, Yang F, et al. Temperature regulates dust activities over the Tibetan Plateau[J]. The Innovation, 2025, 6(4).

---

## Author Comment (AC3)

**Response to the referee #3**

*This manuscript, entitled "On the presence of high nitrite ($NO_2^-$) in coarse particles at Mt. Qomolangma" investigated the soil-derived particulate composition on TP, specifically focused on nitrite, and concluded that this is a major source of HONO on TP. Although I appreciate their work, the manuscript has strong intrinsic weaknesses in the method used in measurement as well as the data interpretations, leaving many large holes in this work. Therefore, this work cannot be accepted in the current form for ACP publication.*

**Response:** We sincerely appreciate the considerable time and effort the referee devoted to reviewing our manuscript. We acknowledge that methodological and data interpretation required clarification and improvement. In response, we have made substantial revisions throughout the manuscript to address the concerns. In the revised version, we have clearly described the field sampling procedures and laboratory analyses. Furthermore, we have improved the data interpretation in accordance with your suggestions, as well as those of the other referees. Please find our detailed, point-by-point responses below. We believe these revisions have significantly improved the scientific rigor and clarity of the manuscript. We respectfully invite you to review the revised manuscript, in which we have carefully addressed all major and minor concerns raised.

*Here are my major remarks.*
*First, let's focus on the methods section. In brief, the sampling method has never been validated in both the lab and field for the TSP nitrite isotopic analysis. The soil extraction method with MQ water is problematic for nitrate and other ions, although it's been proven to be more effective for nitrite. Below are the specific comments for the methods.*

**Response:** Thank you for this helpful suggestion. In the revised manuscript, we have provided a more detailed description regarding the field sampling and laboratory analysis. We would like to emphasize that the filter-based sampling for ambient aerosol and subsequent isotopic analysis of aerosol components is a well-established and widely accepted approach in the scientific community. Numerous peer-reviewed studies have successfully applied this method for isotopic characterization of sulphate, nitrate, ammonium and other water-soluble species in atmospheric particles, and we have followed these established protocols in previous studies. First, the filter-based sampling approach has been widely used for particle nitrite and other water-soluble inorganic ions analysis (i.e., Bhattarai et al., 2023; Bhattarai et al., 2019; Nie et al., 2012; Vernier et al., 2022). After sampling, TSP and $PM_{2.5}$ filter samples were immediately wrapped in pre-baked aluminum foil and stored in frozen until analyzed to minimize potential loss of nitrite. Milli-Q ultrapure water (18.2 MΩ cm) was used to extract the water-soluble inorganic components, and the concentration was determined using ion chromatography. Second, azide method was used for the nitrite isotope analysis following Casciotti et al., 2007. This method has been extensively validated and optimized in our laboratory (Zhou et al., 2022; Zhang et al., 2025), ensuring accuracy and reproducibility of the isotopic measurements in this study.

Regarding the soil ions extraction method: in this study, our goal was to investigate the potential importance of wind-blown dust on the observed high nitrite in TSP. Nitrite in soil was extracted using Milli-Q purewater for the following reasons (1) ultrapure water has been shown to be effective for soil nitrite determination (Homyak et al., 2015); (2) nitrite in TSP and $PM_{2.5}$ samples were also extracted using ultrapure water, ensuring consistency in comparing water-soluble nitrite in surface

soil and aerosol. Moreover, the use of Milli-Q water extraction is a common practice in studies investigating the hygroscopic properties of mineral dust (usually with surface soil as surrogate for laboratory-generated dust aerosol) and the sources of water-soluble ions in dust aerosols in arid and semi-arid environments (Gaston et al., 2017; Tang et al., 2019; Chen et al., 2020; Wu et al., 2022).

*Line 117, the mountain and valley wind can be from all directions. It is hard to say "upwind" or "downwind" in these areas.*

**Response:** Thank you. We agree that using terms like "upwind" or "downwind" can be ambiguous in mountainous regions due to complex topography and variable wind patterns. However, in the Rongbuk Valley, long-term observations have revealed that the local wind system is characterized as a typical katabatic wind pattern, i.e., the strong down-slope wind along the glacier (i.e., the katabatic or glacier wind) begins in the afternoon, reaches the maximum around sunset and maintains until midnight, and then the wind starts from the opposite direction (i.e., up-slope) (Ye and Gao, 1979; Zhu et al., 2006; Zou et al., 2008; Song et al., 2007). In addition, field observations also indicated a prevailing southeasterly wind direction during the springtime campaign (Figure S1 in supporting information). During sampling, to minimize the influence of local anthropogenic activities on sampling, the instruments were strategically set in the southeast (upwind direction) and approximately 100 m away from the living space of the Base Camp. Therefore, we think it makes sense to use the term of "upwind and downwind" here.

[Figure]

Figure S1. The wind direction and wind speed (WS, m s$^{-1}$) at the sampling site during the "Earth Summit Mission-2022" scientific expedition in 2022.

*Line 126, while the authors claim the Whatman quartz filter collects TSP, no test has been done for HONO collection on the quartz filter. There was no method validation.*

**Response:** Thank you. We would like to clarify that our sampling strategy was specifically designed for the collection and analysis of particulate water-soluble ions, including nitrite ($NO_2^-$), and not for gaseous HONO collection. One can't collect HONO with a filter alone, instead in the literature HONO is collected using a denuder system (Chai et al., 2019; Chai et al., 2020). That filters won't collect HONO is also consistent with our observations that the $PM_{2.5}$ filter samples didn't show any detectable nitrite, i.e., if HONO can be collected in filters, we would expect nitrite to be present in both $PM_{2.5}$ and TSP samples. Therefore, we are confident that the detected nitrite originates from particulate matter rather than gaseous HONO artifacts.

*Lack of explanation. Line 130, how is "filter face velocity of approximately 0.288 m s-1." derived?*

**Response:** Thank you for pointing this out. Sorry for the confusion, here we wanted to state the sampling flow (30 L min$^{-1}$). In the revised manuscript, we have deleted this sentence.

*Additionally, details on "surface soil" collection are needed.*

**Response:** Thank you for pointing this out. Additional details on the surface soil sampling have been added to the revised manuscript (lines 152-157):

Surface soil samples (0-5 cm depth, n = 9) were collected in May, 2023 from the east slope, west slope and south sides of the Rongbuk valley. A polytetrafluoroethylene (PTFE) shovel was used to collect soil. The collected soil was immediately transferred to clean plastic bags, sealed and kept frozen. Soil samples were transported into laboratory using a cold chain. Upon arrival at our laboratory, the soil samples were passed through a 60-mesh screen (~0.25 mm) to remove larger particles and thoroughly homogenized prior to chemical and isotopic analysis.

*Lines 182-183, based on the statement here, it is confusing why the authors used the suggested $\Delta^{17}O$ = 0 for N7373 and N23 from Albertin et al., but only use their own measured $\Delta^{17}O$ value of N10219.*

**Response:** Thank you. The two nitrite standards N7373 and N23 are known with zero $\Delta^{17}O$ as measured by different labs, we just used the Albertin et al. 2021 study to support this, i.e., $\Delta^{17}O$ = 0 for the two standards. While for N10219, which is measured with a negative $\Delta^{17}O$ in Albertin et al. 2021, but this value has not been calibrated (note, for zeros one doesn't need to calibrate but for non-zero values calibration are necessary to get the true values). Albertin et al. 2021 just measured N10219 using the azide method and adopted the values of -8.77‰ using the equation of $\Delta^{17}O$ = $\delta^{18}O – 0.52 * \delta^{17}O$ when their measured $\delta^{18}O$ is most close to the true value. Instead, in our lab, as described in our previous publication early this year (Zhang et al., 2025), we developed a technique for measuring $\Delta^{17}O$ of nitrite international standard by oxidation them (and a normal nitrite with $\Delta^{17}O$=0) into nitrate using $O_3$ and corrected the $\Delta^{17}O$ transfer during the oxidation reaction, and adopted (-9.3 ± 0.2) ‰ as the true value. The normal nitrite with $\Delta^{17}O$ = 0 was processed in parallel to quantify $\Delta^{17}O$ transfer during the $O_3$ oxidation of nitrite following Vicars and Savarino (2014). In the revised manuscript, we have made revision to avoid further confusion (lines 207-217): "The $\Delta^{17}O$ values of the three international references have not been certified. To address this, a series laboratory experiments was conducted to determine the true values of three international references in our laboratory (Zhang et al., 2025). In brief, each nitrite international reference was oxidized into $NO_3^-$ by $O_3$ produced from commercial ozone generator. A parallel flow of $O_3$ was also used to convert a normal $KNO_2$ salt ($\Delta^{17}O$=0) into $NO_3^-$ to quantify the $\Delta^{17}O$ transfer during $O_3$ oxidation, following the approach of Vicars and Savarino (2014). Based on these experiments, the $\Delta^{17}O$ of RSIL-N7373 and RSIL-N23 are determined to be negligible, consistent with previous findings (Albertin et al., 2021), while the $\Delta^{17}O$ of RSIL-N10219 is determined to be (-9.3 ± 0.2) ‰ in our laboratory."

*Lines 189-191, although the ion-exchange method has been verified and used for nitrate ion preconcentration for isotopic analysis, no test has been done for nitrite. Before using the method for field samples, is there supposed to be a lab test for different concentrations and different solution environments?*

**Response:** Thank you for pointing out this. The ion exchange resin method follows the same

principle for all soluble ions, which means nitrite and nitrate will all be enriched and then eluted the same. In practice, sure the method needs to be verified. We have done two steps of verification. First, we tested the resin method using nitrite standards, and found there are no significant differences in nitrogen and oxygen isotopes of nitrite standards treated with and without the resin method. Second, for field samples with known amount of nitrite, once the samples were treated by the resin and then measured by mass spectrometer via the azide method, we used the peak sizes of the produced $N_2$ and $O_2$ in the mass spectrometer to estimate the recovery of nitrite using a size calibration curve which is established by repeating measurements of nitrite samples with known amounts varying from 30 nmol to 200 nmol. The result indicates ~100 % yield within analytical uncertainty. In the revised manuscript, we have added the above contents in Supporting Information Text S1 as follows: "The performance of the ion-exchange preconcentration for nitrite isotope analysis was evaluated prior to soil nitrite pretreatment in our laboratory. Briefly, 1 mL of nitrite standards (500 nmol mL$^{-1}$) was diluted to 150 mL and processed following the standard nitrate preconcentration protocol (Erbland et al., 2013). The isotopic analysis demonstrated the ion-exchange method was also effective for nitrite enrichment, with the differences in $\delta^{15}N$ and $\delta^{18}O$ values before and after passage through the ion-exchange resin being less than 1.6‰ across 6 replicates. Moreover, for field samples with known nitrite amount that measured by ion chromatography, the samples were subjected to resin treatment and then analyzed using the mass spectrometry after reduction into $N_2O$ via the azide method. The peak sizes of the resulting $N_2$ and $O_2$ gases were used to estimate nitrite recovery via a calibration curve, which is established by repeating measurements of nitrite samples with known amounts varying from 30 nmol to 200 nmol. Results indicated a recovery rate of approximately 100% within analytical uncertainty."

*Line 199, it is confusing why 6 days was chosen as the HYSPLIY modeling duration. Why and what fire spots need to be identified?*

**Response:** Thank you for this comment. There are growing body of evidence showing that the anthropogenic emissions and biomass burning emissions in South Asia can come across Himalaya and be transported to the TP region (Zhao et al., 2020; Kang et al., 2019; Zhang et al., 2023; Bhattarai et al., 2023; Lin et al., 2021). Accordingly, air mass back trajectory was modelled to assess the potential impact of pollutants from South Asia via long-range transport.

We apologize for the confusion caused by the earlier incorrect statement regarding the HYSPLIT modeling duration. The backward trajectory analysis in our study used a 3-day run time, not 6 days as previously stated. The choice of a 3-day window is consistent with previous studies investigating transboundary pollutant transport from South Asia to the Tibetan Plateau (Lin et al., 2021; Bhattarai et al., 2023). In this study, the modelled air mass trajectories showed that before April 30th, 2022, the air masses generally originated from South Asia and passed through regions with intensive biomass burning activity, generally in agreement with previous reports (Lin et al., 2021; Cong et al., 2015; Bhattarai et al., 2023). To further assess the potential impacts, fire spots along the modeled trajectories were identified using MODIS fire data. This helps evaluate whether biomass burning emissions could contribute to the elevated water-soluble inorganic ions levels in TSP and $PM_{2.5}$ observed at the site before April 30th. We have clarified this rationale in the revised manuscript.

*Lines 164-169, the work used MQ water for soil extraction. While the MQ water may be more effective for nitrite extraction, it has not been proved to be effective for nitrate (and other ions)*

*extraction.*

**Response:** Thank you. In brief, one of our primary objectives was to investigate the potential importance of wind-blown dust on the observed high nitrite in TSP. For this purpose, we prioritized the accurate determination of soil nitrite, for which ultrapure Milli-Q water has been shown to be more effective (Homyak et al., 2015). While we acknowledge that ultrapure water may not be optimal for extracting all ions, including nitrate, our focus was specifically on nitrite. Please also refer to our response to your first comment.

*The authors suggested that the soil-originated TSP nitrite is an important source of atmospheric HONO, but without measuring atmospheric HONO concentration and its isotopic composition. Without seeing the connections between the TSP nitrite and atmospheric HONO in concentration and isotopes, the conclusion hardly makes any sense.*

**Response:** Thank you. But we wanted to first clarify that, we have not concluded "*TSP nitrite is an important source of atmospheric HONO*". In the manuscript, we just speculated that the high level of nitrite in TSP may sever as a potential source of HONO under appropriate conditions which favors partitioning nitrite ion to HONO in the gas phase. However, whether this thermodynamic partitioning process actually occurs under ambient conditions requires further investigation.

But we wanted to note, our speculation is grounded in the well-established thermodynamic gas–particle phase partitioning of semi-volatile species. Given the pKa of nitrous acid (~3.5) and its pH-dependent Henry's law constant, particulate nitrite can dynamically exchange with gaseous HONO in the atmosphere (Park et al., 1988; Chen et al., 2019; Acker et al., 2008). Previous studies have suggested that high levels of particle-phase nitrite can act as a source of atmospheric HONO through the thermodynamic partitioning process (Li et al., 1994; Chen et al., 2019; Wang et al., 2015; Lammel et al., 1988). Our observation of elevated nitrite in TSP, therefore points to a plausible pathway for HONO production. We have revised the manuscript to more clearly state that this is a hypothesis or speculation supported by thermodynamic theory and prior literature, but make no conclusion on this topic.

*Next, the data interpretation for the major statement and conclusion is problematic.*
*Lines 319-328, it is questionable to use the $\delta^{15}N$ isotopic fractionation factor of snow nitrate photolysis to explain the aerosol nitrate photolysis. In order to determine the role of a potential source in the observed isotopic signatures, one would need to use a source apportionment model to quantify the contribution of each source. Simply comparing the $\delta^{15}N$ values between the nitrite and nitrate doesn't yield any meaningful interpretation and can lead to the invalid statement.*

**Response:** Thank you. We agree that applying the $\delta^{15}N$ fractionation factor derived from snow nitrate photolysis to aerosol nitrate may involves uncertainty, as the physical and chemical conditions may differ between snowpack and atmospheric aerosols. However, we note that the photo-induced isotopic fractionation effects during nitrate photolysis in snow and aerosol phase are likely consistent, since in principle it is the difference of zero-point energy (ZPE) between $^{14}NO_3^-$ and $^{15}NO_3^-$ that determines the fractionation effect (Frey et al., 2009; Miller and Yung, 2000). This is why the modeled nitrogen fractionation constant of nitrate in snow by taking the different ZPE (Frey et al., 2009) but no consideration of the snow matrix is very similar, if not identical, to that measured in lab snow nitrate photolysis experiments.

Now back to the statements in our manuscript, the original comparison was intended to refer to existing snow nitrate photolysis fractionation factors as a qualitative framework for interpreting the isotopic difference between nitrite and nitrate observed in our aerosol samples. Theoretical and field studies indicated that $\delta^{15}N$ in $NO_2^-$ from nitrate photolysis will be strongly depleted compared to parent $NO_3^-$ (Frey et al., 2009; Peters et al., 2014). Therefore, the $\delta^{15}N$ similarity between nitrite and nitrate may suggest particulate nitrate photolysis is unlikely the main contributor to the TSP nitrite. Additionally, if particulate nitrate photolysis had played a major role during our sampling campaign, we would expect nitrite to be present in $PM_{2.5}$ samples as well, but not only accumulated in coarse particles. Last, we did not want to do any quantitative assessments here due to the dataset available.

*There is not sufficient quantitative analysis on why HONO uptake and $NO_2$ uptake, and conversion to nitrite pathways are not important. The conclusion that these two pathways are not important was merely based on the assumption or experience that HONO and $NO_2$ concentrations are very low, while the work did not measure these concentrations. As such, this is a very weak conclusion lacking direct evidence. The work did not show any isotopic evidence, even though there is literature available. What is the lifetime of nitrite on coarse particles, and what is the lifetime on fine particles? Why is nitrite more abundant on coarse particles than fine particles? What is the state of the knowledge? There is no explanation on this at all.*

**Response:** Thank you for your detailed and constructive comments. We acknowledge that our initial discussion of the potential roles of atmospheric HONO and $NO_2$ uptake pathways was insufficient supported by quantitative analysis and direct observational evidence. Again, we have to state that in-situ measurements on HONO and NOx were not conducted due to logistical and technical limitations of operating such instrumentation at ~5200 m elevation.

Our conclusion regarding the minor contributions of HONO and $NO_2$ uptake was primarily based on previous field measurements of HONO and $NO_2$ conducted in the background region of central Tibetan Plateau (i.e., Namco station, ~4700 m, from April to June 2019. Wang et al., 2023) and on general atmospheric chemistry knowledge (Ye et al., 2023). The average mixing ratio of HONO at Namco site was 30 pptv, and the average mixing ratio of $NO_2$ was 143 pptv (Wang et al., 2023). Given that our study site of the Base Camp of Mt. Qomolangma is situated at a higher elevation (~ 5200m) and in a more remote environment with fewer anthropogenic and natural sources (i.e., soil microbial activity), one would expect the concentrations of these precursors to be even lower than that observed at Namco (Wang et al., 2023). Under such conditions, the high observed levels of particulate $NO_2^-$ (up to 1300 ng m$^{-3}$) are unlikely to be explained solely by $NO_2$ heterogeneous reactions or HONO uptake. Moreover, if HONO uptake and $NO_2$ heterogeneous reactions occurs, nitrite on fine particles should also be detected due to the large surface area and greater potential for heterogeneous reactions, while our observations indicated $NO_2^-$ only exists in coarse particles. Therefore, we believe our analysis is likely sufficient to rule out their significance in our case.

Moreover, we did not incorporate isotopic measurements (e.g., $\delta^{15}N$ of $NO_2$ or HONO) in our study, although such approaches maybe helpful to identify nitrite formation pathways. Note that the $\delta^{15}N$ isotopes of HONO and $NO_2$, which is a function of the emission sources and N fractionation effect associated with their atmospheric chemistry, may vary significantly among various atmospheric conditions. We agree that incorporating isotopic evidence of concurrent particulate $NO_2^-$, HONO and $NO_2$ would have significantly strengthened our interpretation, and we will investigate this for future research efforts.

Table 2 summarized previous reports on the concentrations of nitrite in TSP and $PM_{2.5}$ samples, and the corresponding formation mechanisms. In this study, we observed nitrite exclusively accumulated in TSP, with no detectable levels on $PM_{2.5}$. If the atmospheric chemistry or processes, such as HONO or $NO_2$ uptake were a dominant formation mechanism, one would expect nitrite to appear in the fine particle fraction, due to their higher surface-area-to-volume ratio and more efficient uptake capacity. The wind-blown dust or biomass burning emissions via long-range transport are more likely responsible for the observed nitrite. The precise mechanisms remain uncertain and merit further investigation through combined field measurements, laboratory experiments, and modeling studies.

*Also, there are several major mistakes in the discussion (Lines 339-347). Specifically, photoenhanced conversion from $NO_2$ to $NO_2^-$ is not "photocatalysis"; $NO_2$ uptake coefficient varies in a significant range, and $1*10^{-5}$ or higher is more typically. For example, see Scharko et al. 2017 ES&T (10.1021/acs.est.7b01363). Also, a typo, "initialed" indicates the recklessness of the manuscript.*

**Response:** Thank you for pointing the mistake out. We have corrected the term "photocatalysis" and now refer to the process more accurately as "photo-enhanced conversion" in the revised manuscript. The typographical error "initialed" has also been corrected to "initiated". In addition, we have thoroughly proofread the manuscript to eliminate similar oversights and improve overall clarity and precision.

Regarding the $NO_2$ uptake coefficient, we agree that it can vary significantly depending on environmental conditions and the nature of the surface. As noted, Scharko et al., 2017 reported $NO_2$ uptake coefficients on soil surface as high as $1\times10^{-5}$, whereas lower values (e.g., $<1\times10^{-6}$) are usually reported on mineral dust or salt surfaces, as summarized in Table 1 of Xuan et al., 2025. Since mineral dust and local soil dominate the TSP composition on the Tibetan Plateau (Kang et al., 2016; Liu et al., 2017; Pokharel et al., 2019), we adopted a lower uptake coefficient in our estimation to reflect the prevailing surface properties. We have clarified this rationale in the revised manuscript and added relevant references to support this choice.

*For the investigation of the influence of biomass burning, the authors only did a single back trajectory for each day, which is not sufficient to determine the relative contributions of different air masses transported from different regions. Furthermore, there is no systematic tracer analysis for biomass burning influences. Therefore, the statement in Lines 371-374 is not valid.*

**Response:** Thank you for this valuable comment. We acknowledge that using a single back trajectory per day may has limitations in fully capturing the variability and relative contributions of different air masses. In supporting information, we have now provided four backward trajectories per day at six-hour intervals to provide a more comprehensive view of air mass origins and improve the robustness of our trajectory analysis (Figure S3). The updated backward trajectories also clearly indicate that before April 30th, 2022, the air masses predominately originated from South Asia and likely bring biomass burning pollutants, while air masses predominately originated from clean regions, such as North and Central TP from May 1st to May 5th, 2022. This shift in air mass source regions is consistent with a significant decrease in potassium concentrations ($K^+$) in $PM_{2.5}$, a common tracer for biomass burning. Specifically, $K^+$ levels declined from $269 \pm 432$ ng m$^{-3}$ (before April 30th) to $22 \pm 12$ ng m$^{-3}$ (after April 30th), with the difference being statistically significant ($p < 0.05$).

Additionally, we agree that the lack of systematic tracer analysis weakens the interpretation regarding biomass burning influences. However, satellite-based fire spot data (MODIS) and $K^+$ in $PM_{2.5}$ indicated the potential impact of biomass burning emissions for samples collected before April $30^{th}$. Furthermore, we have noted this limitation in the revised manuscript and suggested that future studies include more chemical tracers such as levoglucosan to better constrain biomass burning influences.

[Figure]

Figure S3. Similar to Figure 4 in main text. The air-mass backward trajectory was modeled at a 6 h interval each day (panels a-j)

*Additionally, based on lines 375-381, the authors speculate biomass burning smoke won't contribute to the coarse mode particulate nitrite, so why would the authors still track the air mass? Additionally, it is inaccurate to state that "particle nitrite has not yet been detected in biomass burning plumes," according to the cited literature. These works just didn't measure the nitrite concentrations for some reason, one of which is the detection limit, and nitrite is orders of magnitude smaller than nitrate.*

**Response:** Thank you for this insightful comment. We have revised this section to reflect a more consistent interpretation throughout the manuscript. Specially, our results indicated that biomass burning and anthropogenic pollutants from South Asia through long-range atmospheric transport may contribute to the observed TSP nitrite before April $30^{th}$. There are growing body of evidence showing that the anthropogenic emissions and biomass burning emissions in South Asia can come across Himalaya and be transported to the TP region (Zhao et al., 2020; Kang et al., 2019; Zhang et al., 2023 and references therein). The modelled air mass back trajectories further showed that air mass before April $30^{th}$, mainly originated from Nepal and northern India, where extensive fire activities occurred (Figure 3). Therefore, the long-range transport of polluted air could contribute to the observed high water-soluble inorganic ions levels in TSP and $PM_{2.5}$ at Mt. Qomolangma before April $30^{th}$ during the springtime campaign in this study. However, the specific mechanisms responsible for the predominance of nitrite in the coarse particle mode remain unclear,

Moreover, the description regarding "particle nitrite has not yet been detected in biomass burning plumes" was removed accordingly to improve the accuracy of our discussion.

*The interpretation of why nitrite in TSP has a higher $\delta^{18}O$ is invalid. The author states that aerosol water abundance is at least 3 orders of magnitude larger than $NO_2^-$ based only on ISOROPIA modeling results without measurement. As we know, the modeled results are highly uncertain and are greatly influenced by the arbitrary input parameters. More important question— What is the hygroscopic property of the soil-derived TSP? What is the relationship between "aerosol liquid water" and TSP? Is the water really in fine particles or coarse particles? Moreover, if the O exchange between water and nitrite is important in aerosol (again, fine particle or coarse particle?), is it supposed to be important in soils? Finally, what is the explanation for the $\Delta^{17}O$ of soil nitrite compared to the TSP? Unfortunately, there is no clear discussion on this.*

**Response:** Thank you for raising these important points and for the opportunity to clarify our interpretation. We acknowledge the uncertainty associated with estimating aerosol liquid water content (ALWC) solely through thermodynamic modeling. In the absence of direct ALWC measurements, thermodynamic equilibrium models such as ISORROPIA II offer a relatively reliable approach for estimating ALWC. Thermodynamic equilibrium model such as ISORROPIA II have been widely used to predict the aerosol acidity and ALWC (Fountoukis and Nenes, 2007), although the outputs are associated with uncertainty. Note that the performance of ISORROPIA II has been validated in term of water uptake measurements in laboratory experiments over a wide range of atmospherically relevant conditions (Fountoukis and Nenes, 2007). While our study did not include direct measurements of TSP hygroscopicity or aerosol water content, previous research has shown that mineral dust can become hygroscopic once emitted into atmosphere, especially when coated with soluble salts (Nie et al., 2012; Lau et al., 2006; Pathak et al., 2009; Tang et al., 2019; Chen et al., 2020). Moreover, Cardoso et al., 2018 estimated that aerosol water could constituted 20 %–30 % of the total aerosol mass in $PM_{10}$ during dust period through a comprehensive ion mass balance approach. These studies suggested that the aerosol water could also be associated with coarse particles, consistent with the prediction from the thermodynamic equilibrium model.

We agree the oxygen isotopic exchange between nitrite and water molecule also occurs in surface soil, which can explain the significant difference of $\Delta^{17}O$ values between nitrite and nitrate in surface soil (3.7‰ versus 9.6‰). First, an extensive field investigation revealed active denitrification process in dryland surface soil at elevations above 5000m from Tibetan Plateau, accounting for 75% of surface soil nitrite (Wang et al., 2019). Surface soil nitrite produced from denitrification process inherit $\Delta^{17}O$ of nitrate, since this process should follow the mass-dependent fractionation law. Based on the field investigations in Wang et al., 2019, we estimate the surface soil $NO_2^-$ should be associated with relatively higher $\Delta^{17}O$ values (~7.2‰, 9.6‰×0.75) than our determined values (3.7‰). Therefore, we suggest the presence of oxygen atom exchange between nitrite and water molecule in surface soil, which should reduce the $\Delta^{17}O$ of nitrite, may contribute to the relatively low $\Delta^{17}O$ values in surface nitrite. The exchange process maybe particular evident in west slope soil, given the significant lower surface soil $\Delta^{17}O(NO_2^-)$ relative to $\Delta^{17}O(NO_3^-)$ (1.5‰ versus 10.3‰). Note that if the oxygen exchange process was fully efficient, the surface soil $\Delta^{17}O(NO_2^-)$ should be erased to be negligible. Therefore, the determined positive surface soil $\Delta^{17}O(NO_2^-)$ indicated unfavorable conditions for the oxygen isotope exchange process may due to the extremely low soil moisture content (~1%) in surface soil (Ma et al., 2023). Moreover, these hygroscopic transformations of the mineral dust aerosols may enhance water uptake (Tang et al., 2016; Kok et al., 2023), which would facilitate the oxygen exchange process between aerosol nitrite and water molecule. It is suggested that the water-soluble inorganic ions, especially sulfate, nitrate and

ammonium play an important role in the hygroscopic properties of mineral dust aerosol (Lau et al., 2006; Pathak et al., 2009; Tang et al., 2019). In the present study, the sulfate and nitrate account for ~30% of the total mass of water-soluble ions, implying the potential uptake of water vapor on coarse-mode particle surface.

Accordingly, we suggest that the oxygen isotopic exchange between nitrite and water molecule can also explain the observed near-zero TSP $\Delta^{17}O(NO_2^-)$ while soil $\Delta^{17}O(NO_2^-)$ is positive. In the revied manuscript, we added the comparison of $\Delta^{17}O$ between soil nitrite and TSP nitrite according to your and other referee's comments (lines 495-520):

[revised manuscript text omitted]

*Additionally, many vague statements prevent me from understanding the key points significantly. Line 209, what do you mean by "general decline"? Why use 5/1 as the cut-off date? Is there any statistical analysis?*
*Line 212-213, "negligible" is vague.*

*Line 214, what is the point of discussing the $K^+$ in $PM_{2.5}$ that decreases? Why $K^+$ in TSP didn't show the same trend? Instead, the highest $K^+$ occurs after 5/1?*

*Line 215, "comparable" here is not supported by any numerical evidence.*

*Lines 220-222, which figure or summarized data shows "average daytime concentrations of the inorganic species were generally higher than those at night"?*

*Lines 217-218, what does the $Ca^{2+}$ and $Mg^{2+}$ data tell you in addition to "smaller degree"? No point can be achieved.*

*In the paragraph starting with Line 228, the first sentence "The variations of WSIs in TSP generally followed a similar pattern to that in $PM_{2.5}$ (Figure 1)" is an invalid statement as some species have very different trends between TSP and $PM_{2.5}$, such as $K^+$.*

**Response:** Thank you for the insightful and detailed comments. The key point of our findings is the unexpectedly high levels of nitrite, which are predominately accumulated in coarse-mode particulate. We have revised the result section to clarify the points raised and avoid vague or unsupported statements.

**Line 209 ("general decline"):** We have replaced the vague term "general decline" with a more precise description, including quantified changes in concentration levels and corresponding time periods. The cut-off date of April 30th was initially chosen based on observed considerable shifts in atmospheric conditions of water-soluble ions and air mass origins, which we now explain more clearly in the revised text. Additionally, we have included a statistical analysis to support the significance of the observed differences in water-soluble ions before and after this date.

**Line 212-213 ("negligible"):** The term "negligible" has been replaced with specific quantitative values and statistical results to clarify the extent of variation and support the conclusion.

**Line 214 ($K^+$ trends in $PM_{2.5}$ vs TSP):** It is well-documented that biomass bringing emissions in South Asia can come across the Himalayas and contribute significantly to the aerosol loadings over the HTP especially during spring (Bhattarai et al., 2023; Zhao et al., 2020; Kang et al., 2019; Zhang et al., 2023). Out results indicated that there is a substantial decline in water-soluble inorganic ions in $PM_{2.5}$ after April 30th in response to the shifts of air masses origins. Before April 30th, air masses mainly originated from or passed through northern India and Nepal with intensive human activities and numerous fire hotspots, while from May 1st to May 6th the air masses originated from the inside of the TP with rare open fires. Therefore, the time series of $K^+$ in $PM_{2.5}$, a chemical tracer of biomass burning source, was also discussed in this study. In comparison, $K^+$ in TSP may originated from both biomass burning and crustal emissions (Lin et al., 2021), which may explain the different trend of $K^+$ in $PM_{2.5}$ vs TSP during the springtime campaign.

**Line 215 ("comparable"):** We have included the actual numerical data or range of values to support the statement, allowing readers to directly assess the degree of similarity.

**Lines 217–218 ($Ca^{2+}$ and $Mg^{2+}$):** We have revised speculative sentence and focused on data-supported findings.

**Lines 220–222 (daytime versus nighttime concentrations):** We have now referenced the specific figure and provided summarized data (mean ± SD) in the main text or supplemental showing the comparison between daytime and nighttime concentrations for the relevant ions, along with statistical significance where applicable.

**Paragraph starting at Line 228 (TSP vs $PM_{2.5}$ trends):** We agree with the reviewer that the blanket statement was inaccurate. This sentence has been revised to reflect that "Some water-soluble inorganic ions in TSP showed similar variation trends with that in $PM_{2.5}$ throughout the campaign,

while others, such as TSP $K^+$, exhibited divergent behavior". We provide specific examples and refer to the relevant figure (Figure 1 in the main text) for clarity.

This section 3.1 was revised as follows (lines 239-258): "Figure 1 displays the chemical compositions of water-soluble inorganic ions, their corresponding time series and fractional contributions in TSP and $PM_{2.5}$. Throughout the campaign, substantial variations of total WSIs in $PM_{2.5}$ and TSP were observed. For $PM_{2.5}$ samples, the mass concentrations of total WSIs before April 30th were higher than that from May 1st to May 6th ($4.1 \pm 1.7$ versus $1.7 \pm 0.6$ µg m$^{-3}$; $p < 0.05$). The cut-off date of April 30th was initially selected based on observed significantly declines in concentrations of water-soluble ions and the shifts in air mass origins (presented in section 4). This decline of total WSIs after April 30th was predominately driven by significant reductions in secondary inorganic species, i.e., $SO_4^{2-}$, $NO_3^-$ and $NH_4^+$, with the magnitude by more than 60%. In particular, $NH_4^+$ in $PM_{2.5}$ was on average ($322 \pm 243$) ng m$^{-3}$ before April 30th whereas $NH_4^+$ in $PM_{2.5}$ collected during daytime and nighttime of May 1st were 1 ng m$^{-3}$ and 3 ng m$^{-3}$, respectively; and $NH_4^+$ in $PM_{2.5}$ extractions collected from May 2nd to May 6th was below the detection limit. Therefore, the fractional contribution of secondary inorganic species in $PM_{2.5}$ also decreased (Figure 1d). Similarly, $K^+$ in $PM_{2.5}$, a good tracer of biomass burning (Ma et al., 2003), also declined significantly after April 30th ($269 \pm 432$ versus $22 \pm 12$ ng m$^{-3}$; $p < 0.05$). The elevated concentrations of WSIs before April 30th ($4.1 \pm 1.7$ µg m$^{-3}$) are comparable to previous reports at QOMS station ($4.2 \pm 2.2$ µg m$^{-3}$) in the spring (Lin et al., 2021). In comparison, concentrations of $Ca^{2+}$ and $Mg^{2+}$, tracers of wind-blown dust (Wang et al., 2002), decreased by less than 20% after April 30th. In general, $SO_4^{2-}$, $NO_3^-$, and $Ca^{2+}$ are the most abundant species in $PM_{2.5}$, accounting for the majority of the mass of total WSIs. In addition, no clear diurnal variation of water-soluble inorganic ions in $PM_{2.5}$ was observed in this study (Figure S2)."

*Line 239, "similar areas of filters were extracted," doesn't necessarily ensure the accuracy of the comparison because the sampling setups are different.*

**Response:** Thank you for the insightful comment. We acknowledge that differences in sampling design between TSP and $PM_{2.5}$ (e.g., flow rates) may impact the accuracy of direct comparisons. To reduce variability introduced during the extraction process, we used similar filter areas and identical volume of ultrapure water for extraction. It is important to note that, for the same filter area, $PM_{2.5}$ filters generally contained higher aerosol mass loadings than TSP, which likely resulted in higher concentrations of water-soluble inorganic ions in $PM_{2.5}$ extracts. Despite this, nitrite concentrations in $PM_{2.5}$ extracts consistently remained below detection limits, supporting the robustness of our conclusion regarding the absence of nitrite in $PM_{2.5}$.

*Vague statement in Lines 282-283, "High soil $NO_2^-$ and $NO_3^-$ concentrations were observed on the west and east...". "high" here is vague.*

**Response:** Thank you. Revised into: "The soil $NO_2^-$ content on the west slope of Rongbuk Valley (on average 124.7 ng g$^{-1}$) is higher than that observed on the east and south sides (75.3 ng g$^{-1}$ and 48.3 ng g$^{-1}$, respectively). Throughout the text, similar revisions have been made in other parts of the manuscript to ensure clarity and precision. We thank you again.

---

## Author Response (AR2)

**Response to Reviewer Comments #1**

*The authors have addressed most of the concerns I raised earlier and significantly improved the quality of the manuscript. They acknowledged the limitations of their results and implications, which, in my opinion, have strengthened their discussion. After reviewing the revised manuscript, I find this version to be much improved, and I have only a few minor comments:*

**Response:** We sincerely appreciate the considerable time and effort the referee devoted to reviewing our manuscript again. We have revised the manuscript accordingly to address the points raised.

1.*I suggest emphasizing the importance and current knowledge gaps regarding atmospheric nitrite and HONO in the opening paragraph, rather than focusing on glacier retreat. The latter seems less directly relevant to your study than, for example, the observation by Wang et al. of unexpectedly high HONO levels at Namco.*

**Response:** Thank you for this suggestion. We agree that emphasizing the importance and current knowledge gaps related to atmospheric nitrite and HONO would better align with the focus of our study. As such, we have revised the opening paragraph accordingly, particularly referencing the unexpected high HONO levels observed by Wang et al. 2023 at Namco station. The opening paragraph is now revised into:

"The Tibetan Plateau (TP), known as the "Third Pole", represents one of the most climate-sensitive regions on Earth (Yao et al., 2012). Over recent decades, the TP has experienced significant and rapid climate warming, primarily driven by increasing aerosol loadings and greenhouse gas concentrations due to its geographic proximity to East Asia and South Asia with intensive anthropogenic emissions (Kang et al., 2019; Lau et al., 2010; Lüthi et al., 2015). Atmospheric oxidation capacity (AOC) regulates secondary aerosol formation and trace gases removal, including $CH_4$ (Wang et al., 2023; Ye et al., 2023; Ye et al., 2016; Andersen et al., 2023), therefore acting as a critical link between atmospheric pollution and regional climate warming. Previous studies have suggested that strong solar radiation, high ozone ($O_3$) and relatively high water vapor dominate the relatively strong AOC over the TP (Lin et al., 2008). Recent field campaign further highlighted the rapid reactive nitrogen cycling, with N(III) species (i.e., HONO) as the intermediate, also plays an important role in maintaining the strong AOC in TP (Wang et al., 2023). For example, Wang et al. reported high-than-expected HONO (~30 ± 13 pptv) in the Namco station, a representative background site in the central TP, with identified HONO sources including $NO_2$ heterogenous conversion, soil emission and particulate nitrate photolysis (Wang et al., 2023). Incorporating the observed HONO into model simulations approximately doubled the estimated OH abundance compared to simulations without HONO constraints. However, a detailed HONO budget analysis indicated these three dominant sources could not account for the observed daytime HONO levels at the background site, implying the existence of additional, yet unidentified, HONO sources."

2.*Reactions R2 and R3 do not appear to be referenced in the main text. Please consider adding relevant citations or removing them if unnecessary.*

**Response:** Thank you. This is caused by adding/removing references in different versions, we should have been more carefully reviewing these details after each revision. We have removed in the newly submitted revision.

3.*Since you propose that nitrite sources differ before and after May 1—mainly from soil after May 1, and a mix of soil and long-range transport before—Section 4.3 would be more consistent if you compare only the post–May 1 samples to the soil signature. For instance, the $\delta^{15}N$ value of "soil-derived nitrite" should be $-8.0 \pm 0.7‰$ rather than $-7.3 \pm 3.1‰$.*

**Response:** Thank you for this valuable suggestion. The sentence was revised accordingly:

"In addition, the comparable $\delta^{15}N$ values of $NO_2^-$ between TSP collected after May 1st (-8.0 ± 0.7 ‰) and the surface soil (-10.3 ± 3.0‰) also supports that locally resuspended surface soil is an important contributor to the observed high levels of TSP $NO_2^-$."

**Response to Reviewer Comments #3**

*Here is a major question that needs to be addressed. The work reported isotopic signatures of nitrate. Was this nitrate also extracted MQ water? Is it necessarily a complete extraction? If not, is there any fractionation?*

**Response:** We sincerely appreciate the considerable time and effort the referee devoted to reviewing our manuscript again.

For nitrate extraction from the Whatman quartz filters using Milli-Q ultra-pure water, we have tested the extraction efficiency using a double-extraction method: after the initial 20 mL Milli-Q ultra-pure water extraction in an ultrasonic bath for 30 min at room temperature, an additional 20 mL of ultra-pure water was used to re-extract the same filter. The nitrate concentration in the second extract was generally less than 3% of that in the first, indicating that the initial extraction removed nearly all of the nitrate in the filter. Therefore, ultrasonic extraction with Milli-Q ultra-pure water provides near-complete extraction of nitrate from Whatman quartz filters, and thus there should be negligible effects of isotope fractionation.

For soil nitrate isotope analysis, a 2M KCl solution was used (5 g soil with 15 mL 2M KCl) following the method of Fang et al. (2015). The N and O isotopic signatures were determined using the denitrifier method, in which nitrate in the KCl extract is converted to $N_2O$ gas and subsequently analyzed for its isotopic signature. Since the nitrate isotope in surface soil is invoked in this version of manuscript, the method for soil nitrate isotope analysis was included according to your suggestion. Thank you once again for your valuable feedback.

**Reference**

Fang Y, Koba K, Makabe A, et al. Microbial denitrification dominates nitrate losses from forest ecosystems[J]. Proceedings of the National Academy of Sciences, 2015, 112(5): 1470-1474